# Distributed Saddle-Point Problems Under Similarity

**Aleksandr Beznosikov**
MIPT,* HSE University and Yandex, Russia
anbeznosikov@gmail.com

**Gesualdo Scutari**
Purdue University, USA
gscutari@purdue.edu

**Alexander Rogozin**
MIPT and HSE University, Russia
aleksandr.rogozin@phystech.edu

**Alexander Gasnikov**
MIPT, HSE University and ISP RAS,† Russia
gasnikov@yandex.ru

## Abstract

We study solution methods for (strongly-)convex-(strongly)-concave Saddle-Point Problems (SPPs) over networks of two type–master/workers (thus centralized) architectures and mesh (thus decentralized) networks. The local functions at each node are assumed to be *similar*, due to statistical data similarity or otherwise. We establish lower complexity bounds for a fairly general class of algorithms solving the SPP. We show that a given suboptimality $\epsilon > 0$ is achieved over master/workers networks in $\Omega\big(\Delta \cdot \delta/\mu \cdot \log(1/\varepsilon)\big)$ rounds of communications, where $\delta > 0$ measures the degree of similarity of the local functions, $\mu$ is their strong convexity constant, and $\Delta$ is the diameter of the network. The lower communication complexity bound over mesh networks reads $\Omega\big(1/\sqrt{\rho} \cdot \delta/\mu \cdot \log(1/\varepsilon)\big)$, where $\rho$ is the (normalized) eigengap of the gossip matrix used for the communication between neighbouring nodes. We then propose algorithms matching the lower bounds over either types of networks (up to log-factors). We assess the effectiveness of the proposed algorithms on a robust regression problem.

## 1 Introduction

We study smooth (strongly-)convex-(strongly-)concave SPPs over a network of $M$ agents:

$$\min_{x \in X} \max_{y \in Y} f(x,y) := \frac{1}{M} \sum_{m=1}^{M} f_m(x,y), \tag{P}$$

where $X, Y \subseteq \mathbb{R}^d$ are convex and compact sets common to all the agents; and $f_m(x,y)$ is the loss function of agent $m$, known only to the agent. Problem (P) has found a wide range of applications, including, game theory [42, 10], image deconvolution problems [7], adversarial training [3, 12], and statistical learning [1]–see Sec. 2 for some motivating examples in the distributed setting. We are particularly interested in learning problems, where each $f_m$ is the empirical risk that measures the mismatch between the model to be learned and the *local* dataset owned by agent $m$.

Since the functions $f_m$ can be accessed only locally and routing local data to other agents is infeasible or highly inefficient, solving (P) calls for the design of distributed algorithms that alternate between a local computation procedure at each agent's side, and a round of communication among (suitably chosen) neighboring nodes. We address such a design considering explicitly two type of computational architectures, namely: **(i) master/workers** networks–these are centralized systems suitable for parallel computing; for instance, they are the typical computational architecture arising

---

*Moscow Institute of Physics and Technology
†ISP RAS Research Center for Trusted Artificial Intelligence

35th Conference on Neural Information Processing Systems (NeurIPS 2021).

from federated learning applications (e.g., [17]), where data are split across multiple workers and computations are performed in parallel, coordinated by the master node(s); and **(ii) mesh networks**– these are distributed systems with no special topology (modeled just as undirected graphs), which capture scenarios wherein there is no hierarchical structure (e.g., master nodes) and each node can communicate only with its intermediate neighbors.

**Function similarity:** Motivated in particular by machine learning applications, our design and analysis pertain to distributed algorithms for SPPs (P) where the local functions $f_m$'s are *related*– quantities such as gradients and the second derivatives matrices of $f_m$'s differ only by a finite quantity $\delta > 0$; we will term such SPPs as $\delta$-related SPPs. For instance, this is the typical situation in the aforementioned distributed empirical risk minimization setting [2, 14, 47]: when data are i.i.d. among machines, the $f_m$'s reflect statistical similarities in the data residing at different nodes, resulting in a $\delta = \tilde{O}(1/\sqrt{n})$, where $n$ is the local sample size ($\tilde{O}$ hides log-factors and dependence on $d$).

While SPPs have been extensively studied in the centralized setting (e.g., [10, 29, 18, 30, 5]) and more recently over mesh networks [23, 27, 22, 26, 36, 4, 6], we are not aware of any analysis or (distributed) algorithm that explicitly exploit function similarity to boost communication efficiency–either lower complexity bounds or upper bounds. On the other hand, recent works for sum-utility minimization problems over networks (e.g., [2, 38, 35, 45, 43, 11, 47, 14, 39, 20]) show that employing some form of statistical preconditioning in the algorithm design provably reduces communication complexity. Whether these improvements are possible/achievable for $\delta$-related SSPs in the form (P) remains unclear. This paper provides a positive answer to the above open problem.

**Major contributions:** Our major results are summarized next. **(a) Lower complexity bounds:** Under mild structural assumptions on the algorithmic oracle (satisfied by a variety of methods), we establish lower complexity bounds for the $\delta$-related SPP (P) with $\mu$-strongly-convex-strongly-concave, $L$-smooth (twice-differentiable) local functions: an $\varepsilon$ precision on the optimality gap over master/workers system is achieved in $\Omega(\Delta \cdot \delta/\mu \cdot \log(1/\varepsilon))$ communication steps, where $\Delta$ is the diameter of the network. The lower complexity bound over mesh networks reads $\Omega(1/\sqrt{\rho} \cdot \delta/\mu \cdot \log(1/\varepsilon))$ rounds of communications, where $\rho$ is the (normalized) eigengap of the gossip matrix used for the communication between neighbouring nodes. These new lower bounds show a more favorable dependence on the optimization parameters (via $\delta/\mu$) than that of distributed oracles for SPPs ignoring function similarity [5, 36], whose communication complexity, e.g., over mesh networks reads $\Omega(1/\sqrt{\rho} \cdot L/\mu \cdot \log(1/\varepsilon))$. The latter provides a pessimistic prediction when $\delta/\mu \ll L/\mu$. This is the typical situation of ill-conditioned problems, such as many learning problems where the regularization parameter that is optimal for test predictive performance is so small that a scaling with $L/\mu$ is no longer practical while $\delta/\mu$ is (see, e.g., [25, 14]). **(b) Near optimal algorithms:** We proposed algorithms for such SPPs over master/workers and mesh networks that match the lower bounds up to logarithmic factors. They are provably faster than existing solution methods for $\mu$-strongly-convex-strongly-concave, $L$-smooth SPPs, which do not exploit function similarity. Preliminary numerical results on distributed robust logistic regression support our theoretical findings.

## 1.1 Related works

**Methods for SPPs ignoring function similarity:** (Strongly)-convex-(strongly)-concave SPPs have been extensively studied in the optimization literature and as special instances of (strongly) monotone Variational Inequalities (VI) [10, 16]. Several algorithms are available in the **centralized** setting, some directly imported from the VI literature; representative examples include: the mirror-proximal algorithm [29], Extragradient method [18] and the scheme in [30]–they are readily implementable on master/workers architectures as well. For SPPs with $\mu$-strongly-convex-strongly-concave, $L$-smooth loss, all these schemes achieve iteration complexity of $O(L/\mu \cdot \log(1/\varepsilon))$, which has been shown to be optimal for first-order methods solving such a class of SPPs [46, 34]. Lower bounds and optimal algorithms in the **distributed** setting for SPPs without similarity have been studied in [5].

Note that none of the above lower (and upper) complexity bounds or (centralized or distributed) algorithmic designs capture function similarity. As a consequence, convergence rates certified in the aforementioned works, when applicable to $\delta$-related SPPs in the form (P), provide quite pessimistic predictions, in the setting $1 + \delta/\mu \ll L/\mu$.

**Methods for sum-utility minimization exploiting function similarity:** Several works exploited the idea of statistical preconditioning to provably improve communication complexity of solution methods for the minimization of the sum of $\delta$-related, $\mu$-strongly convex and $L$-smooth functions over mas-

ter/workers networks. Lower complexity bounds are established in [2], and read $\Omega\big(\sqrt{\delta/\mu}\log(1/\varepsilon)\big)$, which contrasts with $O\big(\sqrt{L/\mu}\log(1/\varepsilon)\big)$ achievable by first-order (Nesterov) accelerated methods [31], certifying thus faster rates whenever $\delta/\mu < L/\mu$. Solutions methods exploiting function similarity are mirror proximal-like schemes, and include [38, 35, 45] (for quadratic losses), [47] (for self-concordant losses), [43], and [11] (for composite optimization), with [14] employing acceleration. None of these methods are implementable over mesh networks, because they rely on a centralized (master) node. To our knowledge, Network-DANE [20] and SONATA [39] are the only two methods that leverage statistical similarity to enhance convergence of distributed methods over mesh networks; [20] studies strongly convex quadratic losses while [39] considers general objectives, achieving a communication complexity of $\widetilde{O}((1/\sqrt{\rho}) \cdot \delta/\mu \cdot \log(1/\varepsilon))$, where $\widetilde{O}$ hides logarithmic factors. None of the methods above however are applicable to the $\delta$-related SPP (P).

## 1.2 Notation

Given a positive integer $M$, we define $[M] = \{1, \dots, M\}$. We use $\langle x, y \rangle := \sum_{i=1}^{d} x_i y_i$ to denote standard inner product of $x, y \in \mathbb{R}^d$. It induces $\ell_2$-norm in $\mathbb{R}^d$ in the following way $\|x\| := \sqrt{\langle x, x \rangle}$. We also introduce $\mathrm{proj}_{\mathcal{Z}}(z) = \min_{u \in \mathcal{Z}} \|u - z\|$ – the Euclidean projection onto $\mathcal{Z}$. We order the eigenvalues of any symmetrix matrix $A \in \mathcal{R}^{m \times m}$ in nonincreasing fashion, i.e., $\lambda_{\max}(A) = \lambda_1(A) \geq \dots \geq \lambda_m(A) = \lambda_{\min}(A)$, with $\lambda_{\max}(\cdot)$ [resp. $\lambda_{\min}(\cdot)$] denoting the largest (resp. smallest) eigenvalue.

## 2 Setup and Background

**Problem setting:** We begin introducing the main assumptions underlying Problem (P) and some useful notation.

Let us stack the $x$- and $y$-variables in the tuple $z = (x, y)$; accordingly, define $\mathcal{Z} = \mathcal{X} \times \mathcal{Y}$ and the vector-functions $F_m, F : \mathcal{Z} \to \mathcal{R}^{2d}$:

$$F_m(z) := \begin{pmatrix} \nabla_x f_m(x,y) \\ -\nabla_y f_m(x,y) \end{pmatrix}, \quad \text{and} \quad F(z) := \frac{1}{M} \sum_{m=1}^{M} F_m(z). \tag{1}$$

The following conditions are standard for strongly convex-strongly concave SPPs.

**Assumption 1** *Given* (P)*, the following hold:*

*(i)* $\emptyset \neq \mathcal{Z}$ *is a convex set;*

*(ii) Each* $f_m : \mathcal{R}^{2d} \to \mathcal{R}$ *is twice differentiable on (an open set containing)* $\mathcal{Z}$*, with $L$-Lipschitz gradient:* $\|F_m(z_1) - F_m(z_2)\| \leq L\|z_1 - z_2\|$*, for all* $z_1, z_2 \in \mathcal{Z}$*;*

*(iii)* $f(z)$ *is* $\mu$*-strongly convex-strongly concave on* $\mathcal{Z}$*, i.e.,* $\langle F(z_1) - F(z_2), z_1 - z_2 \rangle \geq \mu\|z_1 - z_2\|^2$*, for all* $z_1, z_2 \in \mathcal{Z}$*;*

*(iv) Each* $f_m(z)$ *is convex-concave on* $\mathcal{Z}$*, i.e.* $0$*-strongly convex-strongly concave.*

We are interested in finding the solution $z^* = (x^*, y^*)$ of Problem (P) under function similarity.

**Assumption 2** ($\delta$**-related** $f_m$**'s**) *The local functions are* $\delta$*-related: for all* $(x, y) \in \mathcal{Z}$*,*

$$\|\nabla_{xx}^2 f_m(x,y) - \nabla_{xx}^2 f(x,y)\| \leq \delta,$$
$$\|\nabla_{xy}^2 f_m(x,y) - \nabla_{xy}^2 f(x,y)\| \leq \delta,$$
$$\|\nabla_{yy}^2 f_m(x,y) - \nabla_{yy}^2 f(x,y)\| \leq \delta.$$

The interesting case is when $1 + \delta/\mu \ll L/\mu$. When the $f_m$'s are empirical loss functions over local data sets of size $n$, under standard assumptions on data distributions and learning model (e.g., [47, 14]), $\delta = \tilde{O}(1/\sqrt{n})$ with high probability ($\tilde{O}$ hides log-factors and dependence on $d$)–some motivating examples falling in this category are discussed in Sec. 2.1 below. While such examples

represent important applications, we point out that our (lower and upper) complexity bounds are valid in all scenarios wherein Assumption 2 holds, not necessarily due to statistical arguments.

**Network setting:** The communication network is modeled as a fixed, connected, undirected graph, $\mathcal{G} \triangleq (\mathcal{V}, \mathcal{E})$, where $\mathcal{V} \triangleq \{1, \ldots, M\}$ denotes the vertex set–the set of agents–while $\mathcal{E} \triangleq \{(i, j) \mid i, j \in \mathcal{V}\}$ represents the set of edges–the communication links; $(i, j) \in \mathcal{E}$ iff there exists a communication link between agent $i$ and $j$. We denote by $\Delta$ the diameter of the graph. When it comes to distributed algorithms over mesh networks, we leverage neighbouring communications among adjoining nodes. Communications of $d$-dimensional vectors will be modeled as a matrix multiplication by a matrix $W$ (a.k.a. gossip matrix). The following assumptions on $W$ are standard to establish convergence of distributed algorithms over mesh networks.

**Assumption 3** *The matrix $W \in \mathcal{R}^{M \times M}$ satisfies the following: **(a)** It is compliant with $\mathcal{G}$, that is, (i) $w_{ii} > 0, \forall i \in [M]$; (ii) $w_{ij} > 0$, if $\{j, i\} \in \mathcal{E}$; and (iii) $w_{ij} = 0$ otherwise; **(b)** It is symmetric and stochastic, that is, $W1 = 1$ (and thus also $1^\top W = 1^\top$).*

Notice that a direct consequence of Assumption 3 (along with the fact that $\mathcal{G}$ is connected) is that

$$\rho \triangleq 1 - \max\{\lambda_2(W), |\lambda_{\min}(W)|\} < 1, \tag{2}$$

where $\rho$ is the eigengap between the first and second largest (magnitude) eigenvalue of $W$. Roughly speaking, $\rho$ measures how fast the network mixes information (the larger, the faster).

## 2.1 Motivating examples

Several problems of interest can be cast in the SPP (P), for which function similarity arises naturally, some are briefly discussed next.

**Robust Regression**: Consider the robust instance of the linear regression problem in its Lagrangian form:

$$\min_w \max_r \frac{1}{2N} \sum_{i=1}^N (w^T(x_i + r) - y_i)^2 + \frac{\lambda}{2}\|w\|^2 - \frac{\beta}{2}\|r\|^2, \tag{3}$$

where $w$ are the weights of the model, $\{(x_i, y_i)\}_{i=1}^N$ are pairs of the training data, and $r$ models the noise, and $\lambda$ and $\beta$ are the regularization parameters. Let $n$ be the local sample size (thus $N = nm$). The typical regularization parameter that is optimal for test predictive performance is $\lambda = O(1/\sqrt{N})$. Assuming $\beta$ of the same order of $\lambda$ and invoking function similarity $\delta = O(1/\sqrt{n})$ [25, 14] yield a condition number of the problem $\kappa = \mathcal{O}(\sqrt{m \cdot n})$ while $\delta/\mu = \mathcal{O}(\sqrt{m})$. This implies that first order methods applied to (3) will slowdown as the local sample size $n$ grows. Rate scaling with $\delta/\mu$ would be instead independent on the local sample size.

**Adversarial robustness of neural networks:** Recent works have demonstrated that deep neural networks are vulnerable to adversarial examples—inputs that are almost indistinguishable from natural data and yet classified incorrectly by the network [40, 13]. To improve resistance to a variety of adversarial inputs, a widely studied approach is leveraging robust optimization and formulate the training as saddle-point problem [24, 32]:

$$\min_w \max_r \frac{1}{N} \sum_{i=1}^N l(f(w, x_i + r, y_i)^2 + \frac{\lambda}{2}\|w\|^2 - \frac{\beta}{2}\|r\|^2,$$

where $w$ are the weights of the model, $\{(x_i, y_i)\}_{i=1}^N$ are pairs of the training data, $r$ is the so-called adversarial noise, which models a perturbation in the data, and $\lambda$ and $\beta$ are the regularizers.

**Other optimization problems:** Other instances of the SPP are the (online) transport or Wasserstein Barycenter (WB) problems, see [15, 9]. This representation comes from the dual view of transportation polytope. b) Another example is Lagrangian based optimization problems. For instance, consider the minimization of the sum of loss functions, each one associated to one agent, subject to some (common) constraints. The problem can be equivalently rewritten as a saddle-point problem using Lagrangian multipliers. It is easy to check that if the agents' functions are $\delta$-related, then the resulting saddle-point problem is also so.

# 3 Lower Complexity Bounds

In this section we establish lower complexity bounds for centralized (i.e., master/workers-based) and distributed (gossip-based) algorithms. We begin introducing the back-box procedure describing the class of algorithms these lower bounds pertain to.

## 3.1 Optimization/communication oracle

Our procedure models a fairly general class of (centralized and distributed) algorithms over graphs, whereby nodes perform local computation and communication tasks. Computations at each node are based on linear operations involving current or past iterates, gradients, and vector products with local Hessians and their inverses, as well as solving local optimization problems involving such quantities. During communications, the nodes can share (compatibly with the graph topology) any of the vectors they have computed up until that time. The black-box procedure can be formally describe as follows.

**Definition 1 (Oracle)** *Each agent $m$ has its own local memories $\mathcal{M}_m^x$ and $\mathcal{M}_m^y$ for the $x$- and $y$-variables, respectively–with initialization $\mathcal{M}_m^x = \mathcal{M}_m^y = \{0\}$. $\mathcal{M}_m^x$ and $\mathcal{M}_m^x$ are updated as follows.*

● *Local computation: Between communication rounds, each agent $m$ computes and adds to its $\mathcal{M}_m^x$ and $\mathcal{M}_m^y$ a finite number of points $x, y$, each satisfying*

$$
\begin{aligned}
\alpha x + \beta \nabla_x f_m(x,y) \in span\big\{ x' \,, \; & \nabla_x f_m(x',y'), \\
& (\nabla_{xx}^2 f_m(x'',y'') + D)x' \,, \; (\nabla_{xx}^2 f_m(x'',y'') + D)\nabla_x f_m(x',y') \\
& (\nabla_{xx}^2 f_m(x'',y'') + D)^{-1}x' \,, \; (\nabla_{xx}^2 f_m(x'',y'') + D)^{-1}\nabla_x f_m(x',y'), \\
& (\nabla_{xy}^2 f_m(x'',y''))y' \,, \; (\nabla_{xy}^2 f_m(x'',y''))\nabla_y f_m(x',y') \big\}, \\
\theta y - \varphi \nabla_y f_m(x,y) \in span\big\{ y' \,, \; & \nabla_y f_m(x',y'), \\
& (\nabla_{yy}^2 f_m(x'',y'') + D)y' \,, \; (\nabla_{yy}^2 f_m(x'',y'') + D)\nabla_y f_m(x',y') \\
& (\nabla_{yy}^2 f_m(x'',y'') + D)^{-1}y' \,, \; (\nabla_{yy}^2 f_m(x'',y'') + D)^{-1}\nabla_y f_m(x',y'), \\
& (\nabla_{xy}^2 f_m(x'',y''))^T x' \,, \; (\nabla_{xy}^2 f_m(x'',y''))^T \nabla_x f_m(x',y') \big\},
\end{aligned}
\tag{4}
$$

*for given $x', x'' \in \mathcal{M}_m^x$ and $y', y'' \in \mathcal{M}_m^y$; some $\alpha, \beta, \theta, \varphi \geq 0$ such that $\alpha + \beta > 0$ and $\theta + \varphi > 0$; and $D$ is some diagonal matrix (such that all the inverse matrices exist).*

● *Communication: Based upon communication rounds among neighbouring nodes, $\mathcal{M}_m^x$ and $\mathcal{M}_m^y$ are updated according to*

$$
\mathcal{M}_m^x := span\left\{ \bigcup_{(i,m)\in\mathcal{E}} \mathcal{M}_i^x \right\}, \quad \mathcal{M}_m^y := span\left\{ \bigcup_{(i,m)\in\mathcal{E}} \mathcal{M}_i^y \right\}.
\tag{5}
$$

● *Output: The final global output is calculated as:*

$$
x^K \in span\left\{ \bigcup_{m=1}^M \mathcal{M}_m^x \right\}, \quad y^K \in span\left\{ \bigcup_{m=1}^M \mathcal{M}_m^y \right\}.
$$

The above oracle captures a gamut of existing centralized and distributed algorithms. For instance, local computations model either inexact local solutions–e.g., based on single/multiple steps of gradient or Newton-like updates, which corresponds to setting $\alpha = \theta = 1$ and $\beta = \varphi = 0$–or exact solutions of agents' subproblems (via some subroutine algorithm), corresponding to $\alpha = \theta = 0$ and $\beta = \varphi = 1$. Multiple rounds of computations (resp. communications) can be performed between communication rounds (resp. computation tasks). Notice that the proposed oracle builds on [37, 2] for minimization problems over networks–the former modeling only gradient updates and the latter considering only centralized optimization (master/workers systems).

## 3.2 Lower complexity bounds

We are in the position to state our main results on lower communication complexity–Theorem 1 pertains to algorithms over master/workers systems while Theorem 2 deals with mesh networks.

**Theorem 1** *For any $L, \mu, \delta > 0$ and connected graph $\mathcal{G}$ with diameter $\Delta > 0$, there exist a SPP in the form (P) (satisfying Assumption 1) with $\mathcal{Z} = \mathcal{R}^{2d}$ (where $d$ is sufficiently large), $x^* \neq 0$, $y^* \neq 0$, and local functions $f_m$ being $L$-smooth, $\mu$-strongly-convex-strongly-concave, $\delta$-related (Assumption 2) such that any centralized algorithm satisfying Definition 1 produces the following estimate on the global output $z^K = (x^K, y^K)$ after $K$ communication rounds:*

$$\|z^K - z^*\|^2 = \Omega\left(\exp\left(-\frac{K}{\Delta} \cdot \frac{1}{\frac{1}{8}\sqrt{1 + \left(\frac{\delta}{32\mu}\right)^2} - \frac{1}{8}}\right)\|y^*\|^2\right).$$

**Corollary 1** *In the setting of Theorem 1, the number of communication rounds required to obtain a $\varepsilon$-solution is lower bounded by*

$$\Omega\left(\Delta\left(1 + \frac{\delta}{\mu}\right) \cdot \log\left(\frac{\|y^*\|^2}{\varepsilon}\right)\right). \tag{6}$$

**Theorem 2** *For any $L, \mu, \delta > 0$ and $\rho \in (0; 1]$, there exist a SPP in the form (P) (satisfying Assumption 1) with $\mathcal{Z} = \mathcal{R}^{2d}$(where $d$ is sufficiently large), $x^* \neq 0$, $y^* \neq 0$, and local functions $f_m$ being $L$-smooth, $\mu$-strongly-convex-strongly-concave, $\delta$-related (Assumption 2), and a gossip matrix $W$ over the connected graph $\mathcal{G}$, satisfying Assumption 3 and with eigengap $\rho$, such that any decentralized algorithm satisfying Definition 1 and using the gossip matrix $W$ in the communication steps (5) produces the following estimate on the global output $z^K = (x^K, y^K)$ after $K$ communication rounds:*

$$\|z^K - z^*\|^2 = \Omega\left(\exp\left(-K\sqrt{\rho} \cdot \frac{1}{\frac{1}{20}\sqrt{1 + \left(\frac{\delta}{32\mu}\right)^2} - \frac{1}{20}}\right)\|y^*\|^2\right).$$

**Corollary 2** *In the setting of Theorem 2, the number of communication rounds required to obtain a $\varepsilon$-solution is lower bounded by*

$$\Omega\left(\frac{1}{\sqrt{\rho}}\left(1 + \frac{\delta}{\mu}\right) \cdot \log\left(\frac{\|y^*\|^2}{\varepsilon}\right)\right). \tag{7}$$

These lower complexity bounds show an expected dependence on the optimization parameters and network quantities. Specifically, the number of communications scale proportionally to $\delta/\mu$–this generalizes existing lower bounds [5] that do not account for such similarity, resulting instead in the more pessimistic dependence on $L/\mu$–typically $\delta \leq L$. The network impact is captured by the diameter $\Delta$ of the network for master/workers architectures–$\Delta$ communications steps are required in the worst case to transmit a message between two nodes–and the eigengap $\rho$ of the matrix $W$, when arbitrary graph typologies are consider; $1/\sqrt{\rho}$ can be bounded as $\mathcal{O}(T)$, where $T$ is the largest hitting time of the Markov chain with probability transition matrix $W$ [33]. For instance, for fully connected networks $\Delta = 1/\sqrt{\rho} = 1$ while for star networks $\Delta = 1$ and $1/\sqrt{\rho} = \sqrt{M}$. For general graphs, $1/\sqrt{\rho}$ can be larger than $\Delta$, see [28] for more details. To certify the tightness of the derived lower bounds, the next section designs algorithms that reach such bounds.

## 4 Optimal algorithms

### 4.1 Centralized case (master/workers systems)

Our first optimal algorithm is for SPPs over master/workers architectures or more generally networked systems where a spanning tree (with the root as master node) is preliminary set; it is formally described in Algorithm 1. We assumed w.l.o.g. that the master node owns function $f_1$.

Some insights on the genesis of this method are discussed next.

• Consider for a moment the minimization problem $\min_{x \in X} f(x) := \frac{1}{M} \sum_{m=1}^{M} f_m(x)$, under Assumption 2. Following [38] we can solve it invoking the mirror descent algorithm, which reads

$$x^{k+1} = \arg\min_x \left[ \langle \eta \nabla f(x^k), x \rangle + D_\phi(x, x^k) \right], \tag{8}$$

where $D_\phi(x, y) = \phi(x) - \phi(y) - \langle \nabla \phi(y), x - y \rangle$ is the Bregman divergence, with function $\phi(x) = f_1(x) + \frac{\delta}{2} \|x\|^2$. It is shown that we can take stepsize $\eta = 1$ ([48, 14]). Therefore, (8) can be rewritten as

$$x^{k+1} = \arg\min_x \left[ \frac{1}{\delta} f_1(x) + \frac{1}{2} \left\| x - x^k + \frac{1}{\delta}(\nabla f(x^k) - \nabla f_1(x^k)) \right\|^2 \right]. \tag{9}$$

Noting that in Algorithm 1 $\gamma \sim \frac{1}{\delta}$ (see Appendix B.1), one infers the connection between (9) and the updates in lines 3 (i) and 3 (ii). The extra step as in line 3 (iii) is due to the fact that Algorithm 1 solves a SPP (and not a classical minimization as postulated above): gradient descent-like methods as (8) are not optimal for SPPs; in fact, they might diverge when applied to general convex-concave SPPs. Out approach is then to employ Forward-Backward-Forward algorithms [41] or the Extragradient [18] method, which leads to the step in line 3 (iii).

• Another interpretation of the proposed algorithm comes from looking at Problem (P) as a composite minimization problem, with objective function $h_1(x, y) + h_2(x, y)$, with $h_1(x, y) = f_1(x, y)$ and $h_2(x, y) = \frac{1}{M} \sum_{m=1}^{M} (f_m(x, y) - f_1(x, y))$. The first function $h_1$ is $L$-smooth and convex-concave while $h_2$ is $\delta$-smooth and, in general, non-convex-non-concave. Such type of problems can be solved invoking sliding techniques [19, 36].

---

**Algorithm 1 (Star Min-Max Data Similarity Algorithm)**

    **Parameters:** stepsize $\gamma$, accuracy $e$;
    **Initialization:** Choose $z^0 = (x^0, y^0) \in \mathcal{Z}$, $z_m^0 = z^0$, for all $m \in [M]$;
1: **for** $k = 0, 1, 2, \ldots$ **do**
2:     Each worker $m$ computes $F_m(z^k)$ and sends it to the master;
3:     The master node:
   (i) computes $v^k = z^k - \gamma \cdot \left( F(z^k) - F_1(z^k) \right)$;
   (ii) finds $u^k$, s.t. $\|u^k - \hat{u}^k\|^2 \leq e$, where $\hat{u}^k$ is the solution of:

$$\min_{u_x \in \mathcal{X}} \max_{u_y \in \mathcal{Y}} \left[ \gamma f_1(u_x, u_y) + \frac{1}{2}\|u_x - v_x^k\|^2 - \frac{1}{2}\|u_y - v_y^k\|^2 \right]; \tag{10}$$

   (iii) updates $z^{k+1} = \text{proj}_{\mathcal{Z}} \left[ u^k + \gamma \cdot (F(z^k) - F_1(z^k) - F(u^k) + F_1(u^k)) \right]$ and broadcasts $z^{k+1}$ to the workers
4: **end for**

---

It is not difficult to check that Algorithm 1 is an instance of the oracle introduced in Definition 1. It accommodates either exact solutions of the strongly convex subproblems (10) (corresponding to $e = 0$) or inexact ones (up to tolerance $e > 0$)–the latter can be computed, e.g., using Extragradient method [16], which is optimal in this case.

The communication complexity of the method is proved in the next theorem, which certifies that the proposed algorithm is optimal, i.e., achieves the lower bound (6) on the number of required communications–we refer to Appendix B.1 in the supplementary material for a detailed description of the algorithmic tuning as well as a study of the computational complexity when Extragradient method is employed to solve subproblems (10) (up to a suitably chosen tolerance).

**Theorem 3** *Consider Problem* (P) *under Assumptions 1-2 over a connected graph $\mathcal{G}$ with a master node. Let $\{z^k\}$ be the sequence generated by Algorithm 1 with tuning as described in Appendix B.1 (cf. the supplementary material). Then, given $\varepsilon > 0$, the number of communication rounds for $\|z^k - z^*\|^2 \leq \varepsilon$ is $\mathcal{O}\left( (1 + \delta/\mu) \log(1/\varepsilon) \right)$.*

## 4.2 Distributed case (mesh networks)

We consider now mesh networks. Because of the lack of a master node, each agent $m$ now owns local estimates $u_m$ and $v_m$ of the common variables $u$ and $v$, respectively, which are iteratively updated. At each iteration, a node is selected uniformly at random, which plays the role of the master node, performing thus the update of its own local variables, followed by some rounds of communications via accelerated (inexact) gossip protocols [21, 44]–the latter being instrumental to propagate the updates of the $u, v$-variables and gradients across the network. The algorithm is formally introduced in Algorithm 2, with the accelerated gossip procedure described in Algorithm 3.

---

**Algorithm 2 (Distributed Min-Max Data Similarity Algorithm)**

    **Parameters:** stepsize $\gamma$, accuracy $e, e_0, e_1$, communication rounds $H_0, H_1$;
    **Initialization:** Choose $z^0 = (x^0, y^0) \in \mathcal{Z}$, $z_m^0 = z^0$, for all $m \in [M]$;

1: **for** $k = 0, 1, 2, \ldots$ **do**
2:     `Communications:` $\bar{F}_1^k, \ldots \bar{F}_M^k = \text{AccGossip}(F_1(z_1^k), \ldots F_M(z_M^k); H_0)$;
3:     `Local computations:` Choose an index $m_k \in [M]$ uniformly at random; then node $m_k$

      (i) computes $v_{m_k}^k = z_{m_k}^k - \gamma \cdot (\bar{F}_{m_k}^k - F_{m_k}(z_{m_k}^k))$;

      (ii) finds $\tilde{u}_{m_k}^k$, s.t. $\|\tilde{u}_{m_k}^k - \hat{u}_{m_k}^k\|^2 \le e$, where $\hat{u}_{m_k}^k$ is the solution of:

$$\min_{u_x \in \mathcal{X}} \max_{u_y \in \mathcal{Y}} \left[ \gamma f_{m_k}(u_x, u_y) + \frac{1}{2}\|u_x - v_{x,m_k}^k\|^2 - \frac{1}{2}\|u_y - v_{y,m_k}^k\|^2 \right]; \qquad (11)$$

4:     `Communications:` Run accelerated gossip to propagate $\tilde{u}_{m_k}^k$ and update gradient variables:

$$u_1^k, \ldots u_M^k = M \cdot \text{AccGossip}(0, \ldots, 0, \tilde{u}_{m_k}^k, 0 \ldots, 0; H_1),$$
$$\bar{F}_1^{k+1/2}, \ldots \bar{F}_M^{k+1/2} = \text{AccGossip}(F_1(u_1^k), \ldots F_M(u_M^k); H_0);$$

5:     `Update of` $\tilde{z}_{m_k}$`-variable:` node $m_k$ performs

$$\tilde{z}_{m_k}^{k+1} = \tilde{u}_{m_k}^k + \gamma \cdot (\bar{F}_{m_k}^k - F_{m_k}(z_{m_k}^k) - \bar{F}_{m_k}^{k+1/2} + F_{m_k}(\tilde{u}_{m_k}^k));$$

6:     `Communications:` Run accelerated gossip to propagate $\tilde{z}_{m_k}^{k+1}$:

$$\hat{z}_1^{k+1}, \ldots \hat{z}_M^{k+1} = M \cdot \text{AccGossip}(0 \ldots, 0, \tilde{z}_{m_k}^{k+1}, 0 \ldots, 0; H_1);$$

7:     Each worker update $z_i^{k+1} = \text{proj}_{\mathcal{Z}}\left[\hat{z}_i^{k+1}\right]$;
8: **end for**

---

---

**Algorithm 3 (AccGossip)**

    **Input:** $z_1, \ldots, z_M \in \mathcal{R}^{2d}$, and $H > 0$ (communication rounds);
    **Initialization:** Construct matrix $Z$ with rows $z_1^T, \ldots, z_M^T$; Set
$$Z^{-1} = Z, \quad Z^0 = Z, \quad \text{and} \quad \eta = \frac{1 - \sqrt{1 - \lambda_2^2(W)}}{1 + \sqrt{1 - \lambda_2^2(W)}}.$$
1: **for** $t = 0, 1, 2, \ldots, H$ **do**
2:     $Z^{t+1} = (1 + \eta)WZ^t - \eta Z^{t-1}$,
3: **end for**
    **Output:** Rows of $Z^{H+1}$

---

Convergence of the method is established in Theorem 4 below–we refer to Appendix B.2 in the supplementary material for a detailed description of the algorithmic tuning [choice of the stepsize $\gamma$, precision $e$, numbers of communications rounds $H_0, H_1$, and algorithm to solve (10)].

**Theorem 4** *Consider Problem (P) under Assumptions 1-2 over a connected graph $\mathcal{G}$. Let $\{(z_m^k)_{m \in [M]}\}$ be the sequence generated by Algorithm 2 with tuning as described in Appendix B.2 (cf. the supplementary material) and gossip matrix $W$ satisfying Assumption 3. Then, given $\varepsilon > 0$,*

the number of communication rounds for $\|\bar{z}^k - z^*\|^2 \leq \varepsilon$ reads $\tilde{\mathcal{O}}\left(1/\sqrt{\rho} \cdot \left(1 + \delta/\mu\right) \log^2 \frac{1}{\varepsilon}\right)$, where $\bar{z}^k = \frac{1}{M}\sum_{m=1}^{M} z_m^k$.

While the algorithm achieves the lower bound (7), up to log-factors (which now however depends on $\varepsilon$ as well), there is room for improvements. In fact, selecting only one agent at time performing the updates does not fully exploit the potential computational speedup offered by the networking setting. Also, the use of gossip protocols to propagate the updates of a single agent across the entire network seems to be not quite efficient. Designing alternative distributed algorithms overcoming these limitation is a challenging open problem.

## 5 Numerical Results

We simulate the Robust Linear Regression problem which is defined as

$$\min_w \max_{\|r\| \leq R_r} \frac{1}{2N}\sum_{i=1}^{N}(w^T(x_i + r) - y_i)^2 + \frac{\lambda}{2}\|w\|^2 - \frac{\beta}{2}\|r\|^2. \tag{12}$$

where $w$ are the model weights, $\{x_i, y_i\}_{i=1}^{N}$ is the training dataset, and $r$ is the artificially added noise; we use $\ell_2$-regularization on both $w$ and $r$. We solve the problem over a master/workers topology; we consider a network with 25 workers. We test Algorithm 1 wherein the subproblems (10) at the master node are solved with high accuracy using Extragradient method. A description of the tuning of the algorithm parameters can be found in Appendix C. The algorithms are implemented in Python 3.7[3].

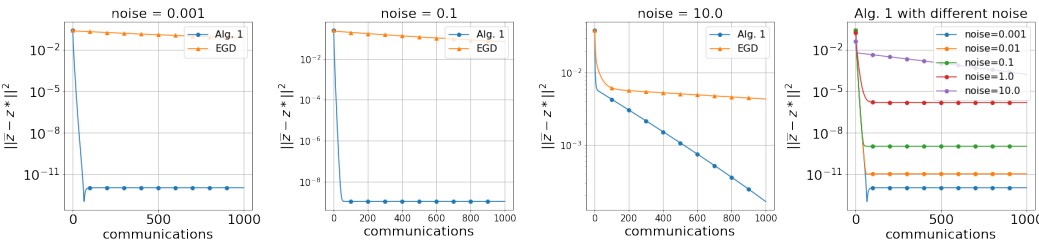

Figure 1: Centralized case, simulated data, 25 workers, ambient dimension $= 40$

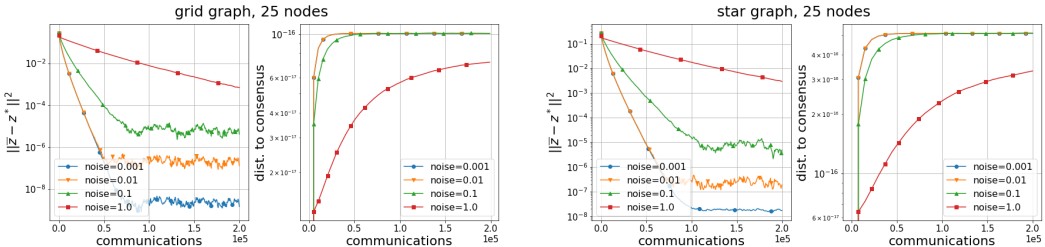

Figure 2: Decentralized case, Alg. 2 with different noise

Our first experiment uses synthetic data, which allows us to control the factor $\delta$, measuring statistical similarity of functions over different nodes. Specifically, we assume all local datasets of size $n = 100$. The data set $\{\hat{x}_i, \hat{y}_i\}_{i=1}^{n}$ at the master node is generated randomly, with each entry of $\hat{x}_i$ and $\hat{y}_i$, $i = 1, \ldots, n$ drawn from the Standard Gaussian distribution. The datasets at the workers' sides, $i = 2, \ldots, M$, are obtained perturbing $\{\hat{x}_i, \hat{y}_i\}_{i=1}^{n}$ by random noise $\xi_i$ with controlled variance.

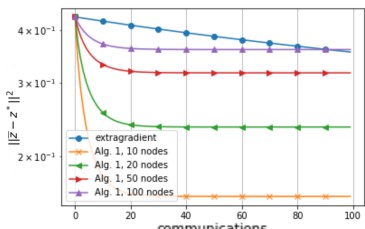

Figure 3: Centralized case, a9a dataset

[3]Source code: https://github.com/alexrogozin12/data_sim_sp

Figure 1 compares the performance of Algorithm 1 and the Centralized Extragradient method [5] applied to Problem (12), under different level of noise added to local datasets (level of similarity), and two different problem and network dimensions – we plot the distance of the iterates from the solution versus the number of communications. It can be seen that Algorithm 1 consistently outperforms the Extragradient method in terms of number of communications–the smaller the noise (the more similar the local functions are), the larger the gap between the two algorithm (in favor of Algorithm 1). On the other hand, at high noise (amplitude 10.0) the performance of Extragradient and Algorithm 1 become comparable. In addition, we compare the performance of Alg.2 under different noise over networks with different topologies in Figure 2.

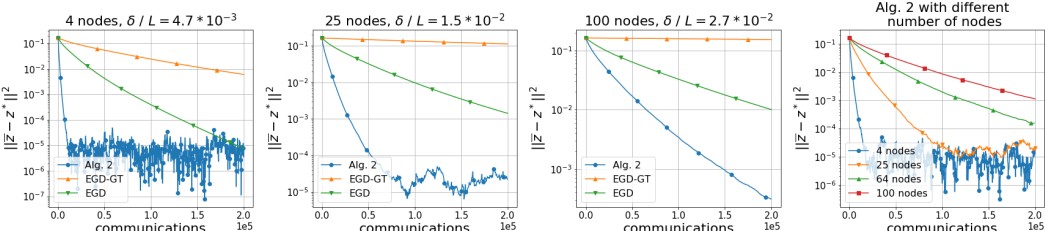

Figure 4: Decentralized case, a9a dataset, grid graph

Our second experiment is using real data, specifically LIBSVM datasets [8]. In this scenario, we do not use additional noise, but still can control the data similarity by choosing the number of workers. The larger the number of workers, the less similar the local functions (less data at each node). Figure 3 compares Algorithm 1 and the Extragradient method: we plot the distance of the iterates from the solution vs. the number of communications. Quite interesting, Algorithm 1 compares favorably even when the number of workers becomes large. Figure 4 compares Algorithm 2 with Decentralized Extragradient method (EGD) [5] and Extragradient method with gradient-tracking (EGD-GT) [27]. The simulations are carried out with parameters tuned according to the theoretical results in the corresponding papers.

## 6    Conclusion

We studied distristributed SPPs over networks, under data similarity. Such problems arise naturally from many applications, including machine learning and signal processing. We first derived lower complexity bounds for such problems for solution methods implementable either on star-networks or on general topologies (modeled as undirected, static graphs). These algorithms are optimal, in the sense that they achieve the lower bounds, up to log factors. The implementation of the proposed method over general network, however, is improvable: by selecting only one agent at time performing the updates, it does not fully exploit the potential computational speedup offered by the parallelism of the networking setting. Also, the use of gossip protocols to propagate the updates of a single agent across the entire network is not very efficient. Another interesting extension would be designing methods that take into account the asymmetry of the function $f$ with respect to the variables $x$ and $y$ (for example, various strong-convexity constants $\mu_x$ and $\mu_y$). Finally, it would be interesting to combine the proposed methods with stochastic/variance reduction techniques to alleviate the cost of local gradient computations.

## Acknowledgments and Disclosure of Funding

The research of A. Rogozin was supported by Russian Science Foundation (project No. 21-71-30005). The work of G. Scutari is supported by the Office of Naval Research, under the Grant # N00014-21-1-2673. The paper was prepared within the framework of the HSE University Basic Research Program.

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
