# Supplementary Material

In this appendix, we provide the proofs of the results presented in the paper; in addition to the case of strongly-convex-strongly-concave functions (discussed therein), here we establish results also for the case of (non strongly) convex-concave functions. In this latter setting, Assumption 1 (iii) (cf. Sec. 2) is fulfilled with $\mu = 0$; in addition, for some $G > 0$ it holds $\|F_m(z^*)\| \leq G$, for all $m$. In the general convex-concave case, we also assume that the set $\mathcal{Z}$ is compact and introduce $\Omega$ – the diameter of $\mathcal{Z}$.

For the sake of convenience, we summarize next the main lower/upper complexity bounds.

| | **lower** | **upper** |
|---|---|---|
| | **centralized** | |
| sc | $\Omega\left(\Delta\left(1+\frac{\delta}{\mu}\right)\log\frac{\|z^0-z^*\|^2}{\varepsilon}\right)$ | $\mathcal{O}\left(\Delta\left(1+\frac{\delta}{\mu}\right)\log\frac{\|z^0-z^*\|^2}{\varepsilon}\right)$ |
| c | $\Omega\left(\Delta\frac{\delta\Omega^2}{\varepsilon}\right)$ | $\mathcal{O}\left(\Delta\frac{\delta\Omega^2}{\varepsilon}\right)$ |
| | **decentralized** | |
| sc | $\Omega\left(\frac{1}{\sqrt{\rho}}\left(1+\frac{\delta}{\mu}\right)\log\frac{\|z^0-z^*\|^2}{\varepsilon}\right)$ | $\tilde{\mathcal{O}}\left(\frac{1}{\sqrt{\rho}}\left(1+\frac{\delta}{\mu}\right)\log^2\frac{\|z^0-z^*\|^2}{\varepsilon}\right)$ |
| c | $\Omega\left(\frac{1}{\sqrt{\rho}}\frac{\delta\Omega^2}{\varepsilon}\right)$ | $\tilde{\mathcal{O}}\left(\frac{1}{\sqrt{\rho}}\frac{\delta\Omega^2}{\varepsilon}\right)$ |

Table 1: Comparison of lower and upper bounds on communication rounds for $\delta$-related smooth strongly-convex–strongly-concave (sc) or convex-concave (c) saddle-point problems in centralized and decentralized cases. Notation: $L$ – smoothness constant of $f_m$, $\mu$ – strongly-convex-strongly-concave constant, $\Omega$ – diameter of optimization set, $\Delta$, $\rho$ – diameter of communication graph and eigengap of the gossip matrix, $\varepsilon$ – precision. In the case of upper bounds for the convex-concave case, the convergence is in terms of the "saddle-point residual" [cf. (16)]; for (sc) functions, it is in terms of the (square) distance to the solution.

## A  Lower Complexity Bounds

We construct the following bilinearly functions with $\delta, \mu$ and $d_x = d_y = d$. Let us consider a linear graph $G$ of $M \geq 3$ nodes. Define $p = \left\lceil \frac{M}{32} \right\rceil$; and let $B = \{1, \ldots p\}$ and $\bar{B} = \{M - p + 1, \ldots, M\}$, with $|B| = |\bar{B}| = p$. The distance in edges $l$ between $B$ and $\bar{B}$ can be bounded by $M - 2p + 1$. We then construct the following bilinear functions on the graph:

$$f_m(x,y) = \begin{cases} f_1(x,y) = \frac{\delta}{4}x^T A_1 y + \frac{p}{M} \cdot 16\mu\|x\|^2 - \frac{p}{M} \cdot 16\mu\|y\|^2 + \frac{\delta^2}{128\mu}e_1^T y, & m \in \bar{B}; \\ f_2(x,y) = \frac{\delta}{4}x^T A_2 y + \frac{p}{M} \cdot 16\mu\|x\|^2 - \frac{p}{M} \cdot 16\mu\|y\|^2, & m \in B; \quad (13) \\ f_3(x,y) = \frac{p}{M} \cdot 16\mu\|x\|^2 - \frac{p}{M} \cdot 16\mu\|y\|^2, & \text{otherwise}; \end{cases}$$

where $e_1 = (1, 0\ldots, 0)$ and

$$A_1 = \begin{pmatrix} 1 & 0 & & & & & \\ & 1 & -2 & & & & \\ & & 1 & 0 & & & \\ & & & 1 & -2 & & \\ & & & & \cdots & \cdots & \\ & & & & & 1 & -2 \\ & & & & & & 1 & 0 \\ & & & & & & & 1 \end{pmatrix}, \quad A_2 = \begin{pmatrix} 1 & -2 & & & & & \\ & 1 & 0 & & & & \\ & & 1 & -2 & & & \\ & & & 1 & 0 & & \\ & & & & \cdots & \cdots & \\ & & & & & 1 & 0 \\ & & & & & & 1 & -2 \\ & & & & & & & 1 \end{pmatrix}.$$

Consider the global objective function:

$$f(x,y) = \frac{1}{M}\sum_{m=1}^{M} f_m(x,y) = \frac{1}{M}\left(|\bar{B}| \cdot f_1(x,y) + |B| \cdot f_2(x,y) + (M - |\bar{B}| - |B|) \cdot f_3(x,y)\right)$$

$$= \frac{2p}{M} \cdot \frac{\delta}{4} x^T A y + \frac{p}{M} \cdot 16\mu\|x\|^2 - \frac{p}{M} \cdot 16\mu\|y\|^2 + \frac{p}{M} \cdot \frac{\delta^2}{128\mu} e_1^T y, \tag{14}$$

with $A = \frac{1}{2}(A_1 + A_2)$.

It is easy to check that

$$\nabla_{xx}^2 f_1(x,y) = \nabla_{xx}^2 f_2(x,y) = \nabla_{xx}^2 f_3(x,y) = \nabla_{xx}^2 f(x,y) = \frac{p}{M} \cdot 16\mu I_x;$$

$$\nabla_{yy}^2 f_1(x,y) = \nabla_{yy}^2 f_2(x,y) = \nabla_{yy}^2 f_3(x,y) = \nabla_{yy}^2 f(x,y) = \frac{p}{M} \cdot 16\mu I_y;$$

$$\nabla_{xy}^2 f_1(x,y) = \frac{\delta}{4} A_1, \quad \nabla_{xy}^2 f_2(x,y) = \frac{\delta}{4} A_2;$$

$$\nabla_{xy}^2 f_3(x,y) = 0, \quad \nabla_{xy}^2 f(x,y) = \frac{2p}{M} \cdot \frac{\delta}{4} A.$$

Note that $f_1, f_2, f_3$ are $L$–smooth (for $L \geq \delta$), $\mu$-strongly-convex–strongly-concave, and $\delta$-related; the last is a consequence of the following

$$\nabla_{xx}^2 f_1(x,y) - \nabla_{xx}^2 f(x,y) = \nabla_{xx}^2 f_2(x,y) - \nabla_{xx}^2 f(x,y) = \nabla_{xx}^2 f_3(x,y) - \nabla_{xx}^2 f(x,y) = 0;$$

$$\nabla_{yy}^2 f_1(x,y) - \nabla_{yy}^2 f(x,y) = \nabla_{yy}^2 f_2(x,y) - \nabla_{yy}^2 f(x,y) = \nabla_{yy}^2 f_3(x,y) - \nabla_{yy}^2 f(x,y) = 0;$$

$$\|\nabla_{xy}^2 f_1(x,y) - \nabla_{xy}^2 f(x,y)\| \leq \|\nabla_{xy}^2 f_1(x,y)\| + \|\nabla_{xy}^2 f(x,y)\| \leq \delta\left(\frac{5}{8} + \frac{p}{M}\right) \leq \delta;$$

$$\|\nabla_{xy}^2 f_2(x,y) - \nabla_{xy}^2 f(x,y)\| \leq \|\nabla_{xy}^2 f_2(x,y)\| + \|\nabla_{xy}^2 f(x,y)\| \leq \delta\left(\frac{5}{8} + \frac{p}{M}\right) \leq \delta;$$

$$\|\nabla_{xy}^2 f_3(x,y) - \nabla_{xy}^2 f(x,y)\| \leq \|\nabla_{xy}^2 f_3(x,y)\| + \|\nabla_{xy}^2 f(x,y)\| \leq \delta\frac{p}{M} \leq \delta.$$

**Lemma 1** *Let Problem* (13) *be solved by any method that satisfies Definition 1. Then after $K$ communication rounds, only the first $\lfloor \frac{K}{l} \rfloor$ coordinates of the global output can be non-zero while the rest of the $d - \lfloor \frac{K}{l} \rfloor$ coordinates are strictly equal to zero. Here $l = M - 2p + 1$ (distance in edges between $B$ and $\bar{B}$).*

**Proof:** We begin introducing some notation, instrumental for our proof. Let

$$E_0 := \{0\}, \quad E_K := \text{span}\{e_1, \ldots, e_K\}.$$

Note that, the initialization reads $\mathcal{M}_m^x = E_0$, $\mathcal{M}_m^y = E_0$.

Suppose that, for some $m$, $\mathcal{M}_m^x = E_K$ and $\mathcal{M}_m^y = E_K$, at some given time. Let us analyze how $\mathcal{M}_m^x, \mathcal{M}_m^y$ can change by performing only local computations.

Firstly, we consider the case when $K$ odd. We have the following:

• For machines $m$ which own $f_1$, it holds

$$\alpha x + \beta A_1 y \in \text{span}\{e_1, x', A_1 y', A_1 A_1^T x'\} = E_K,$$
$$\theta y - \varphi A_1^T x \in \text{span}\{y', A_1^T x', A_1^T A_1 y'\} = E_K.$$

Since $A_1$ has a block diagonal structure with alternating blocks $1 \times 1$ and $2 \times 2$, $A_1^{-1}$ admits the same partitions into $1 \times 1$ and $2 \times 2$ blocks on the diagonal. Therefore, after local computations, we have $\mathcal{M}_m^x = E_K$ and $\mathcal{M}_m^y = E_K$. The situation does not change, no matter how many local computations one does.

• For machines $m$ which own $f_2$, it holds

$$\alpha x + \beta A_2 y \in \text{span}\{x', A_2 y', A_2 A_2^T x'\} = E_{K+1},$$
$$\theta y - \varphi A_2^T x \in \text{span}\{y', A_2^T x', A_2^T A_2 y'\} = E_{K+1},$$

for given $x', x'' \in \mathcal{M}_m^x$ and $y', y'' \in \mathcal{M}_m^y$. It means that, after local computations, one has $\mathcal{M}_m^x = E_{K+1}$ and $\mathcal{M}_m^y = E_{K+1}$. Therefore, machines with function $f_2$ can progress by one new non-zero coordinate.

This means that we constantly have to transfer progress from the group of machines with $f_1$ to the group of machines with $f_2$ and back. Initially, all devices have zero coordinates. Further, machines with $f_1$ can receive the first nonzero coordinate (but only the first, the second is not), and the rest of the devices are left with all zeros. Next, we pass the first non-left coordinate to machines with $f_2$. To do so, $l$ communication rounds are needed. By doing so, they can make the second coordinate non-zero, and then transfer this progress to the machines with $f_1$. Then the process continues in the same way. This completes the proof.

$\square$

The next lemma is devoted to provide an approximate solution of problem (14), and shows that this approximation is close to a real solution. The proof of the lemma follows closely that of [46, Lemma 3.3], and is reported for the sake of completeness.

**Lemma 2 (Lemma 3.3 from [46])** *Let $\alpha = \left(\frac{64\mu}{\delta}\right)^2$ and $q = \frac{1}{2}\left(2 + \alpha - \sqrt{\alpha^2 + 4\alpha}\right) \in (0;1)$–the smallest root of $q^2 - (2+\alpha)q + 1 = 0$; and let define*

$$\bar{y}_i^* = \frac{q^i}{1-q}, \quad i \in [d].$$

*The following bound holds when $\bar{y}^* := [y_1^*, \dots y_d^*]^\top$ is used to approximate the solution $y^*$:*

$$\|\bar{y}^* - y^*\| \leq \frac{q^{d+1}}{\alpha(1-q)}.$$

**Proof:** Let us write the dual function for (14):

$$g(y) = \frac{p}{M} \cdot \left[-\frac{1}{2}y^T\left(\frac{\delta^2}{128\mu}A^T A + 32\mu I\right)y + \frac{\delta^2}{128\mu}e_1^T y\right],$$

where it is not difficult to check that

$$AA^T = \begin{pmatrix} 1 & -1 & & & & & & & \\ -1 & 2 & -1 & & & & & & \\ & -1 & 2 & -1 & & & & & \\ & & -1 & 2 & -1 & & & & \\ & & & -1 & 2 & -1 & & & \\ & & & & & \ddots & & & \\ & & & & & & -1 & 2 & -1 \\ & & & & & & & -1 & 2 \end{pmatrix}.$$

The optimality of dual problem $\nabla g(y^*) = 0$ gives

$$\left(\frac{\delta^2}{128\mu}A^T A + 32\mu I\right)y^* = \frac{\delta^2}{128\mu}e_1,$$

or

$$\left(A^T A + \alpha I\right)y^* = e_1.$$

Equivalently, we can write

$$\begin{cases} (1+\alpha)y_1^* - y_2^* = 1, \\ -y_1^* + (2+\alpha)y_2^* - y_3^* = 0, \\ \dots \\ -y_{d-2}^* + (2+\alpha)y_{d-1}^* - y_d^* = 0, \\ -y_{d-1}^* + (2+\alpha)y_d^* = 0. \end{cases}$$

On the other hand, the approximation $\bar{y}^*$ satisfies the following set of equations:

$$
\begin{cases}
(1 + \alpha)\bar{y}_1^* - \bar{y}_2^* = 1, \\
-\bar{y}_1^* + (2 + \alpha)\bar{y}_2^* - \bar{y}_3^* = 0, \\
\ldots \\
-\bar{y}_{d-2}^* + (2 + \alpha)\bar{y}_{d-1}^* - \bar{y}_d^* = 0, \\
-\bar{y}_{d-1}^* + (2 + \alpha)\bar{y}_d^* = \frac{q^{d+1}}{1-q},
\end{cases}
$$

or equivalently

$$
\left(A^T A + \alpha I\right) \bar{y}^* = e_1 + \frac{q^{d+1}}{1 - q} e_d.
$$

Therefore, the difference between $\bar{y}^*$ and $y^*$ reads

$$
\bar{y}^* - y^* = \left(A^T A + \alpha I\right)^{-1} \frac{q^{d+1}}{1 - q} e_d.
$$

The statement of the lemma follow from the above equality and $\alpha^{-1} I \succeq \left(A^T A + \alpha I\right)^{-1} \succ 0$.

$\square$

The next lemma provides a lower bound for the solution of (14) in the distributed case (13). The proof follows closely that of [46, Lemma 3.4] and is reported for the sake of completeness.

**Lemma 3** *Consider a distributed saddle-point problem with objective function given by (14). For any $K$, choose any problem size $d \geq \max\left\{2 \log_q\left(\frac{\alpha}{4\sqrt{2}}\right), 2K\right\}$, where $\alpha = \left(\frac{64\mu}{\delta}\right)^2$ and $q = \frac{1}{2}\left(2 + \alpha - \sqrt{\alpha^2 + 4\alpha}\right) \in (0; 1)$. Then, any output $\hat{x}, \hat{y}$ produced by any method satisfying Definition 1 after $K$ communications rounds, is such that*

$$
\|\hat{x} - x^*\|^2 + \|\hat{y} - y^*\|^2 \geq q^{\frac{2K}{l}} \frac{\|y_0 - y^*\|^2}{16}.
$$

**Proof:** From Lemma 1 we know that after $K$ communication rounds only $k = \left\lfloor \frac{K}{l} \right\rfloor$ first coordinates in the output can be non-zero. By definition of $\bar{y}^*$, with $q < 1$ and $k \leq \frac{d}{2}$, we have

$$
\begin{aligned}
\|\hat{y} - \bar{y}^*\|^2 &\geq \sqrt{\sum_{j=k+1}^{n} (\bar{y}_j^*)^2} = \frac{q^k}{1-q} \sqrt{q^2 + q^4 + \ldots + q^{2(d-k)}} \\
&\geq \frac{q^k}{\sqrt{2}(1-q)} \sqrt{q^2 + q^4 + \ldots + q^{2n}} = \frac{q^k}{\sqrt{2}} \|\bar{y}^*\|^2 = \frac{q^k}{\sqrt{2}} \|y_0 - \bar{y}^*\|^2.
\end{aligned}
$$

Using Lemma 2 for $d \geq 2 \log_q\left(\frac{\alpha}{4\sqrt{2}}\right)$ we can guarantee that $\bar{y}^* \approx y^*$ (for more detailed proof see [46]) and

$$
\|\hat{x} - x^*\|^2 + \|\hat{y} - y^*\|^2 \geq \|\hat{y} - y^*\|^2 \geq \frac{q^{2k}}{16} \|y_0 - y^*\|^2 = q^{2\lfloor \frac{K}{l} \rfloor} \frac{\|y_0 - y^*\|^2}{16} \geq q^{\frac{2K}{l}} \frac{\|y_0 - y^*\|^2}{16}.
$$

$\square$

## A.1 Centralized case (Theorem 1)

Building on the above preliminary results, we are now ready to prove our complexity lower bound as stated in Theorem 1 of the paper. The following theorem is a more detailed version of the statement in Theorem 1.

**Theorem 5** *Let $L, \mu, \delta > 0$ (with $L > \mu$ and $L > \delta$), $\Delta \in \mathcal{N}$ and $K \in \mathcal{N}$. There exists a centralized saddle-point problem on graph $\mathcal{G}$ for which the following statements are true:*

- *the diameter of graph $G$ is equal to $\Delta$,*

- $f = \frac{1}{M} \sum_{m=1}^{M} f_m : \mathcal{R}^d \times \mathcal{R}^d \to \mathcal{R}$ *are $L$-Lipschitz continuous, $\mu$ – strongly-convex-strongly-concave,*

- $f_m$ *are $L$-Lipschitz continuous, $\mu$ – strongly-convex-strongly-concave, $\delta$-related,*

- *size $d \geq \max\left\{2\log_q\left(\frac{\alpha}{4\sqrt{2}}\right), 2K\right\}$, where $\alpha = \left(\frac{64\mu}{\delta}\right)^2$ and $q = \frac{1}{2}\left(2 + \alpha - \sqrt{\alpha^2 + 4\alpha}\right) \in (0; 1)$,*

- *the solution of the problem is non-zero: $x^* \neq 0$, $y^* \neq 0$.*

*Then for any output $\hat{z}$ of any procedure (Definition 1) with $K$ communication rounds, one can obtain the following estimate:*

$$\|\hat{z} - z^*\|^2 = \Omega\left(\exp\left(-\frac{K}{\Delta} \cdot \frac{1}{\frac{1}{8}\sqrt{1 + \left(\frac{\delta}{32\mu}\right)^2} - \frac{1}{8}}\right)\|y_0 - y^*\|^2\right).$$

**Proof:** It suffices to consider a linear graph with $\Delta + 1$ vertices $\{v_1, \ldots, v_{\Delta+1}\}$ and apply Lemma 1 and Lemma 3. We have

$$\left(\frac{1}{q}\right)^{\frac{2K}{l}} \geq \frac{\|y_0 - y^*\|^2}{16(\|\hat{x} - x^*\|^2 + \|\hat{y} - y^*\|^2)}.$$

Taking the logarithm on both sides, we get

$$\frac{2K}{l} \geq \ln\left(\frac{\|y_0 - y^*\|^2}{16(\|\hat{x} - x^*\|^2 + \|\hat{y} - y^*\|^2)}\right)\frac{1}{\ln(q^{-1})}.$$

Next, we work with

$$\frac{1}{\ln(q^{-1})} = \frac{1}{\ln(1 + (1-q)/q))} = \frac{1 + \frac{\alpha}{2} - \sqrt{\frac{\alpha^2}{4} + \alpha}}{\sqrt{\frac{\alpha^2}{4} + \alpha} - \frac{\alpha}{2}}$$

$$= \frac{\sqrt{\frac{\alpha^2}{4} + \alpha} - \frac{\alpha}{2}}{\alpha} = \sqrt{\frac{1}{4} + \frac{1}{\alpha}} - \frac{1}{2}$$

$$= \sqrt{\frac{1}{4} + \left(\frac{\delta}{64\mu}\right)^2} - \frac{1}{2}.$$

Finally, one can then write

$$\frac{2K}{l} \geq \ln\left(\frac{\|y_0 - y^*\|^2}{16(\|\hat{x} - x^*\|^2 + \|\hat{y} - y^*\|^2)}\right)\left(\frac{1}{2}\sqrt{1 + \left(\frac{\delta}{32\mu}\right)^2} - \frac{1}{2}\right),$$

and

$$\exp\left(\frac{1}{\frac{1}{2}\sqrt{1 + \left(\frac{\delta}{32\mu}\right)^2} - \frac{1}{2}}\frac{2K}{l}\right) \geq \frac{\|y_0 - y^*\|^2}{16(\|\hat{x} - x^*\|^2 + \|\hat{y} - y^*\|^2)},$$

which completes the proof, with $l \geq \frac{1}{2}\Delta$.

$\square$

## A.2   Decentralized case (Theorem 2)

The lower complexity bound as stated in Theorem 2 is proved next. The next theorem is a more detailed version of Theorem 2.

**Theorem 6** *Let $L, \mu, \delta > 0$ (with $L > \mu$ and $L > \delta$), $\rho \in (0;1]$ and $K \in \mathcal{N}$. There exists a distributed saddle-point problem. For which the following statements are true:*

- *a gossip matrix $W$ have $\rho(W) = \rho$,*

- *$f = \frac{1}{M} \sum\limits_{m=1}^{M} f_m : \mathcal{R}^d \times \mathcal{R}^d \to \mathcal{R}$ are $L$-Lipschitz continuous, $\mu$ – strongly-convex-strongly-concave,*

- *$f_m$ are $L$-Lipschitz continuous, $\mu$ – strongly-convex-strongly-concave, $\delta$ - related,*

- *size $d \geq \max\left\{2\log_q\left(\frac{\alpha}{4\sqrt{2}}\right), 2K\right\}$, where $\alpha = \left(\frac{64\mu}{\delta}\right)^2$ and $q = \frac{1}{2}\left(2 + \alpha - \sqrt{\alpha^2 + 4\alpha}\right) \in (0;1)$,*

- *the solution of the problem is non-zero: $x^* \neq 0$, $y^* \neq 0$.*

*Then for any output $\hat{z}$ of any procedure (Definition 1) with $T$ communication rounds, which satisfy Definition 1, one can obtain the following estimate:*

$$\|\hat{z} - z^*\|^2 = \Omega\left(\exp\left(\sqrt{\rho}K \cdot \frac{1}{\frac{1}{20}\sqrt{1 + \left(\frac{\delta}{32\mu}\right)^2} - \frac{1}{20}}\right)\|y_0 - y^*\|^2\right).$$

**Proof:** The proof follow similar steps as in the proof of [37, Theorem 2]. Let $\gamma_M = \frac{1 - \cos\frac{\pi}{M}}{1 + \cos\frac{\pi}{M}}$ be a decreasing sequence of positive numbers. Since $\gamma_2 = 1$ and $\lim_m \gamma_M = 0$, there exists $M \geq 2$ such that $\gamma_M \geq \rho > \gamma_{M+1}$.

- If $M \geq 3$, let us consider linear graph of size $M$ with vertexes $v_1, \dots v_M$, and weighted with $w_{1,2} = 1 - a$ and $w_{i,i+1} = 1$ for $i \geq 2$. Then we applied Lemmas 1 and 3 and get:

$$\|\hat{x} - x^*\|^2 + \|\hat{y} - y^*\|^2 \geq q^{\frac{2K}{l}}\frac{\|y_0 - y^*\|^2}{16}.$$

If $W_a$ is the Laplacian of the weighted graph $\mathcal{G}$, one can note that with $a = 0$, $\rho(W_a) = \gamma_M$, with $a = 1 - \rho(W_a) = 0$. Hence, there exists $a \in (0;1]$ such that $\rho(W_a) = \rho$. Then $\rho \geq \gamma_{M+1} \geq \frac{2}{(M+1)^2}$, and $M \geq \frac{\sqrt{2}}{\sqrt{\rho}} - 1 \geq \frac{1}{4\sqrt{\rho}}$. Finally, $l = M - 2p + 1 \geq \frac{15M}{16} - 1 \geq \frac{15}{16}\left(\frac{\sqrt{2}}{\sqrt{\rho}} - 1\right) - 1 \geq \frac{1}{5\sqrt{\rho}}$ since $\rho \leq \gamma_3 = \frac{1}{3}$. Hence,

$$\|\hat{x} - x^*\|^2 + \|\hat{y} - y^*\|^2 \geq q^{10\sqrt{\rho}K}\frac{\|y_0 - y^*\|^2}{16}.$$

Similarly to the proof of the previous theorem

$$\exp\left(\sqrt{\rho}K \cdot \frac{1}{\frac{1}{20}\sqrt{1 + \left(\frac{\delta}{32\mu}\right)^2} - \frac{1}{20}}\right) \geq \frac{\|y_0 - y^*\|^2}{16(\|\hat{x} - x^*\|^2 + \|\hat{y} - y^*\|^2)}. \tag{15}$$

- If $M = 2$, we construct a totally connected network with 3 nodes with weight $w_{1,3} = a \in [0;1]$. Let $W_a$ is the Laplacian. If $a = 0$, then the network is a linear graph and $\rho(W_a) = \gamma_3 = \frac{1}{3}$. Hence, there exists $a \in [0;1]$ such that $\rho(W_a) = \rho$. Finally, $B = \{v_1\}$, $\bar{B} = \{v_3\}$ and $l \geq 1 \geq \frac{1}{2\sqrt{\rho}}$. Whence it follows that in this case (15) is also valid.

$\square$

## A.3 Regularization and convex-concave case

To establish the lower bounds for the case of (non strongly) convex-concave problems, one can use the classical trick of introducing a regularization and consider instead the following objective function

$$g(x, y) + \frac{\varepsilon}{4\Omega^2} \cdot \|x - x^0\|^2 - \frac{\varepsilon}{4\Omega^2} \cdot \|y - y^0\|^2,$$

which is strongly-convex-strongly-concave with constant $\mu = \frac{\varepsilon}{2\Omega^2}$, where $\varepsilon$ is a precision within the solution of the original problem is computed and $\Omega$ is the diameter of the sets $\mathcal{X}$ and $\mathcal{Y}$. The resulting new SPP problem is solved to $\varepsilon/2$-precision in order to guarantee an accuracy $\varepsilon$ on the solution of the original problem. Therefore, one can directly leverage the lower bound estimates (6) and (7) with the new constants above; this leads to the following lower bounds on the number of communications

$$\Omega\left(\Delta\frac{\delta\Omega^2}{\varepsilon}\right), \qquad \Omega\left(\frac{1}{\sqrt{\rho}}\cdot\frac{\delta\Omega^2}{\varepsilon}\right),$$

for the centralized and decentralized case, respectively.

## B   Optimal algorithms

For the general convex-concave case we introduce the following metric to measure convergence:

$$\text{gap}(z) = \text{gap}(x,y) := \max_{y'\in\mathcal{Y}} f(x,y') - \min_{x'\in\mathcal{X}} f(x',y). \tag{16}$$

### B.1   Centralized case

#### B.1.1   Strongly-convex-strongly-concave case (Proof of Theorem 3)

We begin introducing some intermediate results. Throughout this section, we tacitly subsume all the assumptions as in Theorem 3.

**Lemma 4** *Let $\{z^k\}$ be the sequence generated by Algorithm 1 over $\mathcal{G}$ with a master node. The following holds:*

$$\left\|z^{k+1} - z^*\right\|^2 \le (1-\gamma\mu)\left\|z^k - z^*\right\|^2 - (1 - 3\gamma\mu - 4\gamma^2\delta^2)\left\|z^k - \hat{u}^k\right\|^2$$
$$+ \left(2 + \frac{4\gamma\delta^2}{\mu} + \frac{4}{\gamma\mu} + 4\gamma^2\delta^2\right)\left\|u^k - \hat{u}^k\right\|^2. \tag{17}$$

**Proof:** Define $w^k = u^k + \gamma\cdot(F(z^k) - F_1(z^k) - F(u^k) + F_1(u^k))$. Using the non-expansiveness of the Euclidean projection, we have

$$\left\|z^{k+1} - z^*\right\|^2 = \left\|\text{proj}_\mathcal{Z}\left[w^k\right] - \text{proj}_\mathcal{Z}\left[z^*\right]\right\|^2$$
$$\le \left\|w^k - z^*\right\|^2$$
$$= \left\|z^k - z^*\right\|^2 + 2\langle w^k - z^k, z^k - z^*\rangle + \left\|w^k - z^k\right\|^2$$
$$= \left\|z^k - z^*\right\|^2 + 2\langle w^k - z^k, \hat{u}^k - z^*\rangle + 2\langle w^k - z^k, z^k - \hat{u}^k\rangle + \left\|w^k - z^k\right\|^2$$
$$= \left\|z^k - z^*\right\|^2 + 2\langle w^k - z^k, \hat{u}^k - z^*\rangle + \left\|w^k - \hat{u}^k\right\|^2 - \left\|z^k - \hat{u}^k\right\|^2$$
$$= \left\|z^k - z^*\right\|^2 + 2\langle u^k + \gamma\cdot(F(z^k) - F_1(z^k) - F(u^k) + F_1(u^k)) - z^k, \hat{u}^k - z^*\rangle$$
$$+ \left\|w^k - \hat{u}^k\right\|^2 - \left\|z^k - \hat{u}^k\right\|^2$$
$$= \left\|z^k - z^*\right\|^2 + 2\langle u^k + \gamma\cdot(F(z^k) - F_1(z^k)) - z^k, \hat{u}^k - z^*\rangle$$
$$- 2\gamma\langle F(u^k) - F_1(u^k), \hat{u}^k - z^*\rangle + \left\|w^k - \hat{u}^k\right\|^2 - \left\|z^k - \hat{u}^k\right\|^2.$$

Substituting the expression of $v^k$, we have

$$\left\|z^{k+1} - z^*\right\|^2 \le \left\|z^k - z^*\right\|^2 + 2\langle u^k - v^k, \hat{u}^k - z^*\rangle - 2\gamma\langle F(u^k) - F_1(u^k), \hat{u}^k - z^*\rangle$$
$$+ \left\|w^k - \hat{u}^k\right\|^2 - \left\|z^k - \hat{u}^k\right\|^2$$
$$= \left\|z^k - z^*\right\|^2 + 2\langle \hat{u}^k - v^k, \hat{u}^k - z^*\rangle - 2\gamma\langle F(u^k) - F_1(u^k), \hat{u}^k - z^*\rangle$$
$$+ 2\langle u^k - \hat{u}^k, \hat{u}^k - z^*\rangle + \left\|w^k - \hat{u}^k\right\|^2 - \left\|z^k - \hat{u}^k\right\|^2.$$

Invoking the optimality of $\hat{u}^k$, $\langle \gamma F_1(\hat{u}^k) + \hat{u}^k - v^k, \hat{u}^k - z \rangle \le 0$ (for all $z \in \mathcal{Z}$), yields:

$$
\begin{aligned}
\left\| z^{k+1} - z^* \right\|^2 \le{} & \left\| z^k - z^* \right\|^2 - 2\gamma\langle F_1(\hat{u}^k), \hat{u}^k - z^* \rangle - 2\gamma\langle F(u^k) - F_1(u^k), \hat{u}^k - z^* \rangle \\
& + 2\langle u^k - \hat{u}^k, \hat{u}^k - z^* \rangle + \left\| w^k - \hat{u}^k \right\|^2 - \left\| z^k - \hat{u}^k \right\|^2 \\
={} & \left\| z^k - z^* \right\|^2 - 2\gamma\langle F_1(\hat{u}^k), \hat{u}^k - z^* \rangle - 2\gamma\langle F(\hat{u}^k) - F_1(\hat{u}^k), \hat{u}^k - z^* \rangle \\
& + 2\langle \gamma(F(\hat{u}^k) - F_1(\hat{u}^k) - F(u^k) + F_1(u^k)) + u^k - \hat{u}^k, \hat{u}^k - z^* \rangle \\
& + \left\| w^k - \hat{u}^k \right\|^2 - \left\| z^k - \hat{u}^k \right\|^2. 
\end{aligned}
\tag{18}
$$

Invoking the optimality of the solution $z^*$: $\langle \gamma F(z^*), z^* - z \rangle \le 0$ (for all $z \in \mathcal{Z}$) along with the $\mu$-strong convexity-strong concavity of $f$, we obtain

$$
\begin{aligned}
\left\| z^{k+1} - z^* \right\|^2 \le{} & \left\| z^k - z^* \right\|^2 - 2\gamma\langle F(\hat{u}^k) - F(z^*), \hat{u}^k - z^* \rangle \\
& + 2\langle \gamma(F(\hat{u}^k) - F_1(\hat{u}^k) - F(u^k) + F_1(u^k)) + u^k - \hat{u}^k, \hat{u}^k - z^* \rangle \\
& + \left\| w^k - \hat{u}^k \right\|^2 - \left\| z^k - \hat{u}^k \right\|^2 \\
\le{} & \left\| z^k - z^* \right\|^2 - 2\gamma\mu \left\| \hat{u}^k - z^* \right\|^2 \\
& + 2\langle \gamma(F(\hat{u}^k) - F_1(\hat{u}^k) - F(u^k) + F_1(u^k)) + u^k - \hat{u}^k, \hat{u}^k - z^* \rangle \\
& + \left\| w^k - \hat{u}^k \right\|^2 - \left\| z^k - \hat{u}^k \right\|^2.
\end{aligned}
$$

By Young's inequality, we have

$$
\begin{aligned}
\left\| z^{k+1} - z^* \right\|^2 \le{} & \left\| z^k - z^* \right\|^2 - 2\gamma\mu \left\| \hat{u}^k - z^* \right\|^2 \\
& + \frac{2}{\gamma\mu} \left\| \gamma(F(\hat{u}^k) - F_1(\hat{u}^k) - F(u^k) + F_1(u^k)) + u^k - \hat{u}^k \right\|^2 + \frac{\gamma\mu}{2} \left\| \hat{u}^k - z^* \right\|^2 \\
& + \left\| w^k - \hat{u}^k \right\|^2 - \left\| z^k - \hat{u}^k \right\|^2 \\
\le{} & \left\| z^k - z^* \right\|^2 - \frac{3\gamma\mu}{2} \left\| \hat{u}^k - z^* \right\|^2 \\
& + \frac{4\gamma}{\mu} \left\| F(\hat{u}^k) - F_1(\hat{u}^k) - F(u^k) + F_1(u^k) \right\|^2 + \frac{4}{\gamma\mu} \left\| u^k - \hat{u}^k \right\|^2 \\
& + \left\| u^k + \gamma \cdot (F(z^k) - F_1(z^k) - F(u^k) + F_1(u^k)) - \hat{u}^k \right\|^2 - \left\| z^k - \hat{u}^k \right\|^2 \\
={} & \left\| z^k - z^* \right\|^2 - \frac{3\gamma\mu}{2} \left\| \hat{u}^k - z^* \right\|^2 \\
& + \frac{4\gamma}{\mu} \left\| F(\hat{u}^k) - F_1(\hat{u}^k) - F(u^k) + F_1(u^k) \right\|^2 + \frac{4}{\gamma\mu} \left\| u^k - \hat{u}^k \right\|^2 \\
& + 2 \left\| u^k - \hat{u}^k \right\|^2 + 2\gamma^2 \left\| F(z^k) - F_1(z^k) - F(u^k) + F_1(u^k) \right\|^2 - \left\| z^k - \hat{u}^k \right\|^2.
\end{aligned}
$$

Note that the function $f - f_1$ is $\delta$-smooth, since $\left\| \nabla_{xx}f - \nabla_{xx}f_1 \right\|^2 \le \delta$, $\left\| \nabla_{xy}f - \nabla_{xy}f_1 \right\|^2 \le \delta$, $\left\| \nabla_{yy}f - \nabla_{yy}f_1 \right\|^2 \le \delta$; therefore,

$$
\begin{aligned}
\left\| z^{k+1} - z^* \right\|^2 \le{} & \left\| z^k - z^* \right\|^2 - \frac{3\gamma\mu}{2} \left\| \hat{u}^k - z^* \right\|^2 \\
& + \frac{4\gamma\delta^2}{\mu} \left\| u^k - \hat{u}^k \right\|^2 + \frac{4}{\gamma\mu} \left\| u^k - \hat{u}^k \right\|^2 \\
& + 2 \left\| u^k - \hat{u}^k \right\|^2 + 2\gamma^2\delta^2 \left\| z^k - u^k \right\|^2 - \left\| z^k - \hat{u}^k \right\|^2 \\
\le{} & \left\| z^k - z^* \right\|^2 - \frac{3\gamma\mu}{2} \left\| \hat{u}^k - z^* \right\|^2 - (1 - 4\gamma^2\delta^2) \left\| z^k - \hat{u}^k \right\|^2 \\
& + \left( 2 + \frac{4\gamma\delta^2}{\mu} + \frac{4}{\gamma\mu} + 4\gamma^2\delta^2 \right) \left\| u^k - \hat{u}^k \right\|^2.
\end{aligned}
$$

Finally, using $\left\| a + b \right\|^2 \ge \frac{2}{3} \left\| a \right\|^2 - 2 \left\| b \right\|^2$, we obtain the desired result (17).

$\square$

**Theorem 7** *Let $\{z^k\}$ the sequence generated by Algorithm 1 (in the setting of Theorem 3) with the step-size $\gamma$ given by*

$$\gamma = \min\left\{\frac{1}{12\mu}, \frac{1}{4\delta}\right\}. \tag{19}$$

*Let each subproblem (10) be solved up to (relative) precision $\tilde{e}$,*

$$\tilde{e} = \frac{1}{2\left(2 + \frac{4\gamma\delta^2}{\mu} + \frac{4}{\gamma\mu} + 4\gamma^2\delta^2\right)}. \tag{20}$$

*Then, $\left\|z^K - z^*\right\|^2 \le \varepsilon$ after*

$$K = \mathcal{O}\left(\left(1 + \frac{\delta}{\mu}\right)\log\frac{\left\|z^0 - z^*\right\|^2}{\varepsilon}\right) \quad iterations/communications. \tag{21}$$

**Proof:** The output $u^k$ produced by inner method satisfies

$$\left\|u^k - \hat{u}^k\right\|^2 \le \tilde{e}\left\|z^k - \hat{u}^k\right\|^2.$$

Combining this fact and Lemma 4 yields

$$\left\|z^{k+1} - z^*\right\|^2 \le (1 - \gamma\mu)\left\|z^k - z^*\right\|^2 - (1 - 3\gamma\mu - 4\gamma^2\delta^2)\left\|z^k - \hat{u}^k\right\|^2$$
$$+ \left(2 + \frac{4\gamma\delta^2}{\mu} + \frac{4}{\gamma\mu} + 4\gamma^2\delta^2\right)\tilde{e}\left\|z^k - \hat{u}^k\right\|^2$$
$$\overset{(20)}{\le} (1 - \gamma\mu)\left\|z^k - z^*\right\|^2 - \left(\frac{1}{2} - 3\gamma\mu - 4\gamma^2\delta^2\right)\left\|z^k - \hat{u}^k\right\|^2.$$

The proof is completed by choosing $\gamma$ according to (19).

$\square$

**Corollary 3** *Let we solve the subproblem (10) via Extragradient method with starting point $z^k$ and*

$$T = \mathcal{O}\left((1 + \gamma L)\log\frac{1}{\tilde{e}}\right) \tag{22}$$

*iterations. Then we can estimate the total number local iterations at the server side by*

$$\mathcal{O}\left(\left(1 + \frac{\delta}{\mu} + \frac{L}{\mu}\right)\log\frac{1}{\tilde{e}}\log\frac{\left\|z^0 - z^*\right\|^2}{\varepsilon}\right).$$

**Proof:** Firstly, one can note that after $T$ iterations of Extragradient method from (22) we can achieve $\tilde{e}$ precision. It follows readily from the convergence of Extragradient method [5] and the fact that the objective function in (10) is 1-strongly-convex-strongly-concave and $(1 + \gamma L)$-smooth. Then we can estimate the total number of local iterations at the server side, namely:

$$K \cdot T = \mathcal{O}\left(\frac{1}{\gamma\mu}(1 + \gamma L)\log\frac{1}{\tilde{e}}\log\frac{\left\|z^0 - z^*\right\|^2}{\varepsilon}\right)$$
$$= \mathcal{O}\left(\left(\frac{1}{\gamma\mu} + \frac{L}{\mu}\right)\log\frac{1}{\tilde{e}}\log\frac{\left\|z^0 - z^*\right\|^2}{\varepsilon}\right)$$
$$= \mathcal{O}\left(\left(1 + \frac{\delta}{\mu} + \frac{L}{\mu}\right)\log\frac{1}{\tilde{e}}\log\frac{\left\|z^0 - z^*\right\|^2}{\varepsilon}\right).$$

$\square$

**Remark.** If the server is located in the center of a graph with a diameter $\Delta$, then an additional factor $\Delta$ will appear in the total number of communications (21).

### B.1.2 Convex-Concave case

**Lemma 5** *For one iteration of Algorithm 1, the following estimate holds:*

$$2\gamma\langle F(u^k), u^k - z\rangle \leq \left\|z^k - z\right\|^2 - \left\|z^{k+1} - z\right\|^2 - \left(1 - 2\gamma^2\delta^2\right)\left\|z^k - u^k\right\|^2$$
$$+ (8\gamma L\Omega + 6\gamma G + 2\Omega)\left\|u^k - \hat{u}^k\right\| + 2\left\|u^k - \hat{u}^k\right\|^2. \qquad (23)$$

**Proof:** The proof follows similar steps as that of Lemma 4, with the difference that $z^*$ therein is replaced here with any $z \in \mathcal{Z}$. Specifically, recalling the first equality in (18), we have

$$\left\|z^{k+1} - z\right\|^2 \leq \left\|z^k - z\right\|^2 - 2\gamma\langle F_1(\hat{u}^k), \hat{u}^k - z\rangle - 2\gamma\langle F(u^k) - F_1(u^k), \hat{u}^k - z\rangle$$
$$+ 2\langle u^k - \hat{u}^k, \hat{u}^k - z\rangle + \left\|w^k - \hat{u}^k\right\|^2 - \left\|z^k - \hat{u}^k\right\|^2$$
$$= \left\|z^k - z\right\|^2 - 2\gamma\langle F_1(u^k), u^k - z\rangle - 2\gamma\langle F(u^k) - F_1(u^k), u^k - z\rangle$$
$$+ 2\gamma\langle F_1(u^k) - F_1(\hat{u}^k), u^k - z\rangle + 2\gamma\langle F_1(\hat{u}^k), u^k - \hat{u}^k\rangle$$
$$+ 2\gamma\langle F(u^k) - F_1(u^k), u^k - \hat{u}^k\rangle + 2\langle u^k - \hat{u}^k, \hat{u}^k - z\rangle$$
$$+ \left\|w^k - \hat{u}^k\right\|^2 - \left\|z^k - \hat{u}^k\right\|^2.$$

Small rearrangement gives

$$2\gamma\langle F(u^k), u^k - z\rangle \leq \left\|z^k - z\right\|^2 - \left\|z^{k+1} - z\right\|^2$$
$$+ 2\gamma\langle F_1(u^k) - F_1(\hat{u}^k), u^k - z\rangle + 2\gamma\langle F_1(\hat{u}^k), u^k - \hat{u}^k\rangle$$
$$+ 2\gamma\langle F(u^k) - F_1(u^k), u^k - \hat{u}^k\rangle + 2\langle u^k - \hat{u}^k, \hat{u}^k - z\rangle$$
$$+ \left\|w^k - \hat{u}^k\right\|^2 - \left\|z^k - \hat{u}^k\right\|^2$$
$$\leq \left\|z^k - z\right\|^2 - \left\|z^{k+1} - z\right\|^2$$
$$+ 2\gamma\|F_1(u^k) - F_1(\hat{u}^k)\| \cdot \|u^k - z\| + 2\gamma\|F_1(\hat{u}^k)\| \cdot \|u^k - \hat{u}^k\|$$
$$+ 2\gamma\|F(u^k) - F_1(u^k)\| \cdot \|u^k - \hat{u}^k\| + 2\|u^k - \hat{u}^k\| \cdot \|\hat{u}^k - z\|$$
$$+ \left\|w^k - \hat{u}^k\right\|^2 - \left\|z^k - \hat{u}^k\right\|^2.$$

Invoking the definition of $w^k = u^k + \gamma \cdot (F(z^k) - F_1(z^k) - F(u^k) + F_1(u^k))$, we get

$$2\gamma\langle F(u^k), u^k - z\rangle \leq \left\|z^k - z\right\|^2 - \left\|z^{k+1} - z\right\|^2$$
$$+ 2\gamma\|F_1(u^k) - F_1(\hat{u}^k)\| \cdot \|u^k - z\| + 2\gamma\|F_1(\hat{u}^k)\| \cdot \|u^k - \hat{u}^k\|$$
$$+ 2\gamma\|F(u^k) - F_1(u^k)\| \cdot \|u^k - \hat{u}^k\| + 2\|u^k - \hat{u}^k\| \cdot \|\hat{u}^k - z\|$$
$$+ \left\|u^k + \gamma \cdot (F(z^k) - F_1(z^k) - F(u^k) + F_1(u^k)) - \hat{u}^k\right\|^2 - \left\|z^k - \hat{u}^k\right\|^2$$
$$\leq \left\|z^k - z\right\|^2 - \left\|z^{k+1} - z\right\|^2$$
$$+ 2\gamma\|F_1(u^k) - F_1(\hat{u}^k)\| \cdot \|u^k - z\| + 2\gamma\|F_1(\hat{u}^k)\| \cdot \|u^k - \hat{u}^k\|$$
$$+ 2\gamma\|F(u^k) - F_1(u^k)\| \cdot \|u^k - \hat{u}^k\| + 2\|u^k - \hat{u}^k\| \cdot \|\hat{u}^k - z\|$$
$$+ 2\left\|u^k - \hat{u}^k\right\|^2 + 2\gamma^2\left\|F(z^k) - F_1(z^k) - F(u^k) + F_1(u^k)\right\|^2$$
$$- \left\|z^k - \hat{u}^k\right\|^2.$$

Then we use smoothness of $f - f_1$, $f$, $f_1$ and obtain

$$2\gamma\langle F(u^k), u^k - z\rangle \leq \left\|z^k - z\right\|^2 - \left\|z^{k+1} - z\right\|^2$$
$$+ 2\gamma L\|u^k - \hat{u}^k\| \cdot \Omega + 2\gamma(G + L\Omega) \cdot \|u^k - \hat{u}^k\|$$
$$+ 4\gamma(G + L\Omega)\| \cdot \|u^k - \hat{u}^k\| + 2\Omega \cdot \|u^k - \hat{u}^k\|$$
$$+ 2\left\|u^k - \hat{u}^k\right\|^2 + 2\gamma^2\delta^2\left\|z^k - u^k\right\|^2 - \left\|z^k - \hat{u}^k\right\|^2$$
$$= \left\|z^k - z\right\|^2 - \left\|z^{k+1} - z\right\|^2 - \left(1 - 2\gamma^2\delta^2\right)\left\|z^k - u^k\right\|^2$$
$$+ (8\gamma L\Omega + 6\gamma G + 2\Omega)\|u^k - \hat{u}^k\| + 2\left\|u^k - \hat{u}^k\right\|^2.$$

$$\square$$

Here we additionally used the diameter $\Omega$ of $\mathcal{Z}$ and simple fact:

$$\|F_1(\hat{u}^k)\| - G \leq \|F_1(\hat{u}^k)\| - \|F_1(z^*)\| \leq \|F_1(\hat{u}^k) - F_1(z^*)\| \leq L\Omega. \tag{24}$$

**Theorem 8** *Let problem (10) be solved by Extragradient with precision e:*

$$e = \min\left\{\frac{\varepsilon}{\delta}; \frac{\varepsilon^2}{(L\Omega + G + \delta\Omega)^2}\right\} \tag{25}$$

*and number of iterations $T$:*

$$T = \mathcal{O}\left((1 + \gamma L)\log\frac{\Omega^2}{e}\right).$$

*Additionally, let us choose stepsize $\gamma$ as follows*

$$\gamma = \frac{1}{2\delta}. \tag{26}$$

*Then it holds that $gap(z_{avg}^K) \sim \varepsilon$ after*

$$K = \mathcal{O}\left(\frac{\delta\Omega^2}{\varepsilon}\right) \quad \text{iterations,} \tag{27}$$

*where $z_{avg}^K$ define as follows: $x_{avg}^K = \frac{1}{K}\sum_{k=0}^K u_x^k,\ y_{avg}^K = \frac{1}{K}\sum_{k=0}^K u_y^k$.*

**Proof:** Summing (23) over all $k$ from 0 to $K$

$$2\gamma\sum_{k=0}^K \langle F(u^k), u^k - z\rangle \leq \|z^0 - z\|^2 - (1 - 2\gamma^2\delta^2)\sum_{k=0}^K \|z^k - u^k\|^2$$

$$+ (8\gamma L\Omega + 6\gamma G + 2\Omega)\sum_{k=0}^K \|u^k - \hat{u}^k\| + 2\sum_{k=0}^K \|u^k - \hat{u}^k\|^2.$$

Then, by $x_{avg}^K = \frac{1}{K}\sum_{k=0}^K u_x^k$ and $y_{avg}^K = \frac{1}{K}\sum_{k=0}^K u_y^k$, Jensen's inequality and convexity-concavity of $f$:

$$gap(z_{avg}^K) \leq \max_{y'\in\mathcal{Y}} f\left(\frac{1}{K}\left(\sum_{k=0}^K u_x^k\right), y'\right) - \min_{x'\in\mathcal{X}} f\left(x', \frac{1}{K}\left(\sum_{k=0}^K u_y^k\right)\right)$$

$$\leq \max_{y'\in\mathcal{Y}} \frac{1}{K}\sum_{k=0}^K f(u_x^k, y') - \min_{x'\in\mathcal{X}} \frac{1}{K}\sum_{k=0}^K f(x', u_y^k).$$

Given the fact of linear independence of $x'$ and $y'$:

$$gap(z_{avg}^K) \leq \max_{(x',y')\in\mathcal{Z}} \frac{1}{K}\sum_{k=0}^K \left(f(x^K, y') - f(x', u_y^k)\right).$$

Using convexity and concavity of the function $f$:

$$gap(z_{avg}^K) \leq \max_{(x',y')\in\mathcal{Z}} \frac{1}{K}\sum_{k=0}^K \left(f(u_x^k, y') - f(x', u_y^k)\right)$$

$$= \max_{(x',y')\in\mathcal{Z}} \frac{1}{K}\sum_{k=0}^K \left(f(u_x^k, y') - f(u_x^k, u_y^k) + f(u_x^k, u_y^k) - f(x', u_y^k)\right)$$

$$\leq \max_{(x',y')\in\mathcal{Z}} \frac{1}{K}\sum_{k=0}^K \left(\langle\nabla_y f(u_x^k, u_y^k), y' - u_y^k\rangle + \langle\nabla_x f(u_x^k, u_y^k), u_x^k - x'\rangle\right)$$

$$\leq \max_{z\in\mathcal{Z}} \frac{1}{K}\sum_{k=0}^K \langle F(u^k), u^k - z\rangle.$$

Then it gives with our choice of $\gamma$

$$\text{gap}(z_{avg}^K) \leq \max_{z \in \mathcal{Z}} \frac{\|z^0 - z\|^2}{2\gamma K} + \frac{(4\gamma L\Omega + 3\gamma G + \Omega)}{\gamma K} \sum_{k=0}^K \|u^k - \hat{u}^k\| + \frac{1}{\gamma K} \sum_{k=0}^K \|u^k - \hat{u}^k\|^2$$

$$\leq \frac{\Omega^2}{2\gamma K} + \left(4L\Omega + 3G + \frac{\Omega}{\gamma}\right)\sqrt{e} + \frac{1}{\gamma}e$$

$$= \frac{\delta\Omega^2}{K} + \left(4L\Omega + 3G + 2\delta\Omega\right)\sqrt{e} + 2\delta e.$$

$e$ from (25) is completed the proof.

$\square$

**Remark.** (27) also corresponds to the number of communication rounds. It is also easy to estimate the total number of local iterations on server:

$$K \times T = \mathcal{O}\left(\frac{\delta\Omega^2}{\varepsilon}(1 + \gamma L)\log\frac{\Omega^2}{e}\right)$$

$$= \mathcal{O}\left(\frac{\delta\Omega^2}{\varepsilon}\left(1 + \frac{L}{\delta}\right)\log\frac{\Omega^2}{e}\right)$$

$$= \mathcal{O}\left(\frac{(L+\delta)\Omega^2}{\varepsilon}\log\frac{\Omega^2}{e}\right).$$

## B.2 Decentralized case

Before moving on to the proofs of the decentralized case, let us understand the `AccGossip` convergence [21, 44]:

**Lemma 6** *Assume that $\{y_m\}_{m=1}^M$ are output of Algorithm 3 with input $\{x_m\}_{m=1}^M$. Then it holds that*

$$\sum_{m=1}^M \|y_m - \bar{y}\|^2 \leq (1 - \sqrt{\rho})^{2H}\left(\sum_{m=1}^M \|x_m - \bar{x}\|^2\right). \tag{28}$$

*And $\bar{x} = \frac{1}{M}\sum_{m=1}^M x_m = \frac{1}{M}\sum_{m=1}^M y_m = \bar{y}$.*

From this lemma it holds that for any $i$

$$\|y_i - \bar{y}\|^2 \leq (1 - \sqrt{\rho})^{2H}\left(\sum_{m=1}^M \|x_m - \bar{x}\|^2\right). \tag{29}$$

and

$$\|y_i - \bar{y}\| \leq (1 - \sqrt{\rho})^H\sqrt{\left(\sum_{m=1}^M \|x_m - \bar{x}\|^2\right)}. \tag{30}$$

### B.2.1 Strongly-convex-strongly-concave case

**Lemma 7** *For one iteration of Algorithm 2, the following estimate holds:*

$$\left\|z_{m_k}^{k+1} - z^*\right\|^2 \leq (1 - \gamma\mu)\left\|z_{m_k}^k - z^*\right\|^2 - (1 - 3\gamma\mu - 12\gamma^2\delta^2)\left\|z_{m_k}^k - \hat{u}_{m_k}^k\right\|^2$$

$$+ \left(2 + 12\gamma^2\delta^2 + \frac{4}{\gamma\mu} + \frac{8\gamma\delta^2}{\mu}\right)\left\|\tilde{u}_{m_k}^k - \hat{u}_{m_k}^k\right\|^2$$

$$+ 6\gamma^2\left\|\bar{F}_{m_k}^k - F(z_{m_k}^k)\right\|^2 + \left(6\gamma^2 + \frac{8\gamma}{\mu}\right)\left\|\bar{F}_{m_k}^{k+1/2} - F(\tilde{u}_{m_k}^k)\right\|^2$$

$$+ 2\langle \hat{z}_{m_k}^{k+1} - \tilde{z}_{m_k}^{k+1}, \tilde{z}_{m_k}^{k+1} - z^*\rangle + \left\|\tilde{z}_{m_k}^{k+1} - \hat{z}_{m_k}^{k+1}\right\|^2.$$

**Proof:** Using non-expansiveness of the Euclidean projection, we get

$$
\begin{aligned}
\left\|z_{m_k}^{k+1} - z^*\right\|^2 &= \left\|\mathrm{proj}_{\mathcal{Z}}\left[\hat{z}_{m_k}^{k+1}\right] - \mathrm{proj}_{\mathcal{Z}}\left[z^*\right]\right\|^2 \\
&\leq \left\|\hat{z}_{m_k}^{k+1} - z^*\right\|^2 \\
&= \left\|\tilde{z}_{m_k}^{k+1} - z^*\right\|^2 + 2\langle \hat{z}_{m_k}^{k+1} - \tilde{z}_{m_k}^{k+1}, \tilde{z}_{m_k}^{k+1} - z^*\rangle + \left\|\tilde{z}_{m_k}^{k+1} - \hat{z}_{m_k}^{k+1}\right\|^2 \\
&= \left\|z_{m_k}^{k} - z^*\right\|^2 + 2\langle \tilde{z}_{m_k}^{k+1} - z_{m_k}^{k}, z_{m_k}^{k} - z^*\rangle + \left\|\tilde{z}_{m_k}^{k+1} - z_{m_k}^{k}\right\|^2 \\
&\quad + 2\langle \hat{z}_{m_k}^{k+1} - \tilde{z}_{m_k}^{k+1}, \tilde{z}_{m_k}^{k+1} - z^*\rangle + \left\|\tilde{z}_{m_k}^{k+1} - \hat{z}_{m_k}^{k+1}\right\|^2 \\
&= \left\|z_{m_k}^{k} - z^*\right\|^2 + 2\langle \tilde{z}_{m_k}^{k+1} - z_{m_k}^{k}, \hat{u}_{m_k}^{k} - z^*\rangle + 2\langle \tilde{z}_{m_k}^{k+1} - z_{m_k}^{k}, z_{m_k}^{k} - \hat{u}_{m_k}^{k}\rangle \\
&\quad + \left\|\tilde{z}_{m_k}^{k+1} - z_{m_k}^{k}\right\|^2 + 2\langle \hat{z}_{m_k}^{k+1} - \tilde{z}_{m_k}^{k+1}, \tilde{z}_{m_k}^{k+1} - z^*\rangle + \left\|\tilde{z}_{m_k}^{k+1} - \hat{z}_{m_k}^{k+1}\right\|^2 \\
&= \left\|z_{m_k}^{k} - z^*\right\|^2 + 2\langle \tilde{z}_{m_k}^{k+1} - z_{m_k}^{k}, \hat{u}_{m_k}^{k} - z^*\rangle + \left\|\tilde{z}_{m_k}^{k+1} - \hat{u}_{m_k}^{k}\right\|^2 - \left\|z_{m_k}^{k} - \hat{u}_{m_k}^{k}\right\|^2 \\
&\quad + 2\langle \hat{z}_{m_k}^{k+1} - \tilde{z}_{m_k}^{k+1}, \tilde{z}_{m_k}^{k+1} - z^*\rangle + \left\|\tilde{z}_{m_k}^{k+1} - \hat{z}_{m_k}^{k+1}\right\|^2 \\
&= \left\|z_{m_k}^{k} - z^*\right\|^2 \\
&\quad + 2\langle \tilde{u}_{m_k}^{k} + \gamma \cdot (\bar{F}_{m_k}^{k} - F_{m_k}(z_{m_k}^{k}) - \bar{F}_{m_k}^{k+1/2} + F_{m_k}(\tilde{u}_{m_k}^{k})) - z_{m_k}^{k}, \hat{u}_{m_k}^{k} - z^*\rangle \\
&\quad + \left\|\tilde{z}_{m_k}^{k+1} - \hat{u}_{m_k}^{k}\right\|^2 - \left\|z_{m_k}^{k} - \hat{u}_{m_k}^{k}\right\|^2 \\
&\quad + 2\langle \hat{z}_{m_k}^{k+1} - \tilde{z}_{m_k}^{k+1}, \tilde{z}_{m_k}^{k+1} - z^*\rangle + \left\|\tilde{z}_{m_k}^{k+1} - \hat{z}_{m_k}^{k+1}\right\|^2 \\
&= \left\|z_{m_k}^{k} - z^*\right\|^2 \\
&\quad + 2\langle \tilde{u}_{m_k}^{k} + \gamma \cdot (\bar{F}_{m_k}^{k} - F_{m_k}(z_{m_k}^{k})) - z_{m_k}^{k}, \hat{u}_{m_k}^{k} - z^*\rangle \\
&\quad - 2\gamma\langle \bar{F}_{m_k}^{k+1/2} - F_{m_k}(\tilde{u}_{m_k}^{k})), \hat{u}_{m_k}^{k} - z^*\rangle + \left\|\tilde{z}_{m_k}^{k+1} - \hat{u}_{m_k}^{k}\right\|^2 - \left\|z_{m_k}^{k} - \hat{u}_{m_k}^{k}\right\|^2 \\
&\quad + 2\langle \hat{z}_{m_k}^{k+1} - \tilde{z}_{m_k}^{k+1}, \tilde{z}_{m_k}^{k+1} - z^*\rangle + \left\|\tilde{z}_{m_k}^{k+1} - \hat{z}_{m_k}^{k+1}\right\|^2.
\end{aligned}
$$

Substituting the expression for $v_{m_k}^{k}$, we have

$$
\begin{aligned}
\left\|z_{m_k}^{k+1} - z^*\right\|^2 &\leq \left\|z_{m_k}^{k} - z^*\right\|^2 + 2\langle \tilde{u}_{m_k}^{k} - v_{m_k}^{k}, \hat{u}_{m_k}^{k} - z^*\rangle \\
&\quad - 2\gamma\langle \bar{F}_{m_k}^{k+1/2} - F_{m_k}(\tilde{u}_{m_k}^{k})), \hat{u}_{m_k}^{k} - z^*\rangle + \left\|\tilde{z}_{m_k}^{k+1} - \hat{u}_{m_k}^{k}\right\|^2 - \left\|z_{m_k}^{k} - \hat{u}_{m_k}^{k}\right\|^2 \\
&\quad + 2\langle \hat{z}_{m_k}^{k+1} - \tilde{z}_{m_k}^{k+1}, \tilde{z}_{m_k}^{k+1} - z^*\rangle + \left\|\tilde{z}_{m_k}^{k+1} - \hat{z}_{m_k}^{k+1}\right\|^2 \\
&= \left\|z_{m_k}^{k} - z^*\right\|^2 + 2\langle \hat{u}_{m_k}^{k} - v_{m_k}^{k}, \hat{u}_{m_k}^{k} - z^*\rangle \\
&\quad - 2\gamma\langle \bar{F}_{m_k}^{k+1/2} - F_{m_k}(\tilde{u}_{m_k}^{k})), \hat{u}_{m_k}^{k} - z^*\rangle + \left\|\tilde{z}_{m_k}^{k+1} - \hat{u}_{m_k}^{k}\right\|^2 - \left\|z_{m_k}^{k} - \hat{u}_{m_k}^{k}\right\|^2 \\
&\quad + 2\langle \tilde{u}_{m_k}^{k} - \hat{u}_{m_k}^{k}, \hat{u}_{m_k}^{k} - z^*\rangle + 2\langle \hat{z}_{m_k}^{k+1} - \tilde{z}_{m_k}^{k+1}, \tilde{z}_{m_k}^{k+1} - z^*\rangle + \left\|\tilde{z}_{m_k}^{k+1} - \hat{z}_{m_k}^{k+1}\right\|^2.
\end{aligned}
$$

According to the optimal condition for $\hat{u}_{m_k}^{k}$: $\langle \gamma F_{m_k}(\hat{u}_{m_k}^{k}) + \hat{u}_{m_k}^{k} - v_{m_k}^{k}, \hat{u}_{m_k}^{k} - z\rangle \leq 0$ (for all $z \in \mathcal{Z}$),

$$
\begin{aligned}
\left\|z_{m_k}^{k+1} - z^*\right\|^2 &\leq \left\|z_{m_k}^{k} - z^*\right\|^2 - 2\gamma\langle F_{m_k}(\hat{u}_{m_k}^{k}), \hat{u}_{m_k}^{k} - z^*\rangle \\
&\quad - 2\gamma\langle \bar{F}_{m_k}^{k+1/2} - F_{m_k}(\tilde{u}_{m_k}^{k}), \hat{u}_{m_k}^{k} - z^*\rangle + \left\|\tilde{z}_{m_k}^{k+1} - \hat{u}_{m_k}^{k}\right\|^2 - \left\|z_{m_k}^{k} - \hat{u}_{m_k}^{k}\right\|^2 \\
&\quad + 2\langle \tilde{u}_{m_k}^{k} - \hat{u}_{m_k}^{k}, \hat{u}_{m_k}^{k} - z^*\rangle + 2\langle \hat{z}_{m_k}^{k+1} - \tilde{z}_{m_k}^{k+1}, \tilde{z}_{m_k}^{k+1} - z^*\rangle + \left\|\tilde{z}_{m_k}^{k+1} - \hat{z}_{m_k}^{k+1}\right\|^2 \\
&= \left\|z_{m_k}^{k} - z^*\right\|^2 - 2\gamma\langle F_{m_k}(\hat{u}_{m_k}^{k}), \hat{u}_{m_k}^{k} - z^*\rangle \\
&\quad - 2\gamma\langle F(\hat{u}_{m_k}^{k}) - F_{m_k}(\hat{u}_{m_k}^{k}), \hat{u}_{m_k}^{k} - z^*\rangle \\
&\quad - 2\gamma\langle \bar{F}_{m_k}^{k+1/2} - F(\hat{u}_{m_k}^{k}) - F_{m_k}(\tilde{u}_{m_k}^{k}) + F_{m_k}(\hat{u}_{m_k}^{k}), \hat{u}_{m_k}^{k} - z^*\rangle \\
&\quad + \left\|\tilde{z}_{m_k}^{k+1} - \hat{u}_{m_k}^{k}\right\|^2 - \left\|z_{m_k}^{k} - \hat{u}_{m_k}^{k}\right\|^2 \\
&\quad + 2\langle \tilde{u}_{m_k}^{k} - \hat{u}_{m_k}^{k}, \hat{u}_{m_k}^{k} - z^*\rangle + 2\langle \hat{z}_{m_k}^{k+1} - \tilde{z}_{m_k}^{k+1}, \tilde{z}_{m_k}^{k+1} - z^*\rangle + \left\|\tilde{z}_{m_k}^{k+1} - \hat{z}_{m_k}^{k+1}\right\|^2.
\end{aligned}
$$

Applying property of the solution $z^*$: $\langle \gamma F(z^*), z^* - z \rangle \leq 0$ (for all $z \in \mathcal{Z}$). And then $\mu$-strong convexity - strong concavity of $f$, we obtain

$$
\begin{aligned}
\left\| z_{m_k}^{k+1} - z^* \right\|^2 \leq {}& \left\| z_{m_k}^k - z^* \right\|^2 - 2\gamma \langle F(\hat{u}_{m_k}^k) - F(z^*), \hat{u}_{m_k}^k - z^* \rangle \\
& - 2\gamma \langle \bar{F}_{m_k}^{k+1/2} - F(\hat{u}_{m_k}^k) - F_{m_k}(\tilde{u}_{m_k}^k) + F_{m_k}(\hat{u}_{m_k}^k), \hat{u}_{m_k}^k - z^* \rangle \\
& + \left\| \tilde{z}_{m_k}^{k+1} - \hat{u}_{m_k}^k \right\|^2 - \left\| z_{m_k}^k - \hat{u}_{m_k}^k \right\|^2 \\
& + 2\langle \tilde{u}_{m_k}^k - \hat{u}_{m_k}^k, \hat{u}_{m_k}^k - z^* \rangle + 2\langle \hat{z}_{m_k}^{k+1} - \tilde{z}_{m_k}^{k+1}, \tilde{z}_{m_k}^{k+1} - z^* \rangle + \left\| \tilde{z}_{m_k}^{k+1} - \hat{z}_{m_k}^{k+1} \right\|^2 \\
\leq {}& \left\| z_{m_k}^k - z^* \right\|^2 - 2\gamma\mu \left\| \hat{u}_{m_k}^k - z^* \right\|^2 \\
& - 2\gamma \langle \bar{F}_{m_k}^{k+1/2} - F(\hat{u}_{m_k}^k) - F_{m_k}(\tilde{u}_{m_k}^k) + F_{m_k}(\hat{u}_{m_k}^k), \hat{u}_{m_k}^k - z^* \rangle \\
& + \left\| \tilde{z}_{m_k}^{k+1} - \hat{u}_{m_k}^k \right\|^2 - \left\| z_{m_k}^k - \hat{u}_{m_k}^k \right\|^2 \\
& + 2\langle \tilde{u}_{m_k}^k - \hat{u}_{m_k}^k, \hat{u}_{m_k}^k - z^* \rangle + 2\langle \hat{z}_{m_k}^{k+1} - \tilde{z}_{m_k}^{k+1}, \tilde{z}_{m_k}^{k+1} - z^* \rangle + \left\| \tilde{z}_{m_k}^{k+1} - \hat{z}_{m_k}^{k+1} \right\|^2.
\end{aligned}
$$

By Young's inequality, we have

$$
\begin{aligned}
\left\| z_{m_k}^{k+1} - z^* \right\|^2 \leq {}& \left\| z_{m_k}^k - z^* \right\|^2 - 2\gamma\mu \left\| \hat{u}_{m_k}^k - z^* \right\|^2 \\
& + \frac{4\gamma}{\mu} \left\| \bar{F}_{m_k}^{k+1/2} - F(\hat{u}_{m_k}^k) - F_{m_k}(\tilde{u}_{m_k}^k) + F_{m_k}(\hat{u}_{m_k}^k) \right\|^2 \\
& + \frac{\gamma\mu}{4} \left\| \hat{u}_{m_k}^k - z^* \right\|^2 + \left\| \tilde{z}_{m_k}^{k+1} - \hat{u}_{m_k}^k \right\|^2 - \left\| z_{m_k}^k - \hat{u}_{m_k}^k \right\|^2 \\
& + \frac{4}{\gamma\mu} \left\| \tilde{u}_{m_k}^k - \hat{u}_{m_k}^k \right\|^2 + \frac{\gamma\mu}{4} \left\| \hat{u}_{m_k}^k - z^* \right\|^2 \\
& + 2\langle \hat{z}_{m_k}^{k+1} - \tilde{z}_{m_k}^{k+1}, \tilde{z}_{m_k}^{k+1} - z^* \rangle + \left\| \tilde{z}_{m_k}^{k+1} - \hat{z}_{m_k}^{k+1} \right\|^2 \\
= {}& \left\| z_{m_k}^k - z^* \right\|^2 - \frac{3\gamma\mu}{2} \left\| \hat{u}_{m_k}^k - z^* \right\|^2 \\
& + \frac{4\gamma}{\mu} \left\| \bar{F}_{m_k}^{k+1/2} - F(\hat{u}_{m_k}^k) - F_{m_k}(\tilde{u}_{m_k}^k) + F_{m_k}(\hat{u}_{m_k}^k) \right\|^2 \\
& + \left\| \tilde{u}_{m_k}^k + \gamma \cdot (\bar{F}_{m_k}^k - F_{m_k}(z_{m_k}^k) - \bar{F}_{m_k}^{k+1/2} + F_{m_k}(\tilde{u}_{m_k}^k)) - \hat{u}_{m_k}^k \right\|^2 \\
& - \left\| z_{m_k}^k - \hat{u}_{m_k}^k \right\|^2 + \frac{4}{\gamma\mu} \left\| \tilde{u}_{m_k}^k - \hat{u}_{m_k}^k \right\|^2 \\
& + 2\langle \hat{z}_{m_k}^{k+1} - \tilde{z}_{m_k}^{k+1}, \tilde{z}_{m_k}^{k+1} - z^* \rangle + \left\| \tilde{z}_{m_k}^{k+1} - \hat{z}_{m_k}^{k+1} \right\|^2 \\
\leq {}& \left\| z_{m_k}^k - z^* \right\|^2 - \frac{3\gamma\mu}{2} \left\| \hat{u}_{m_k}^k - z^* \right\|^2 \\
& + \frac{8\gamma}{\mu} \left\| F(\tilde{u}_{m_k}^k) - F(\hat{u}_{m_k}^k) - F_{m_k}(\tilde{u}_{m_k}^k) + F_{m_k}(\hat{u}_{m_k}^k) \right\|^2 \\
& + \frac{8\gamma}{\mu} \left\| \bar{F}_{m_k}^{k+1/2} - F(\tilde{u}_{m_k}^k) \right\|^2 \\
& + 6\gamma^2 \left\| F(z_{m_k}^k) - F_{m_k}(z_{m_k}^k) - F(\tilde{u}_{m_k}^k) + F_{m_k}(\tilde{u}_{m_k}^k) \right\|^2 \\
& + 6\gamma^2 \left\| \bar{F}_{m_k}^k - F(z_{m_k}^k) \right\|^2 + 6\gamma^2 \left\| \bar{F}_{m_k}^{k+1/2} - F(\tilde{u}_{m_k}^k) \right\|^2 \\
& + 2 \left\| \tilde{u}_{m_k}^k - \hat{u}_{m_k}^k \right\|^2 - \left\| z_{m_k}^k - \hat{u}_{m_k}^k \right\|^2 \\
& + \frac{4}{\gamma\mu} \left\| \tilde{u}_{m_k}^k - \hat{u}_{m_k}^k \right\|^2 + 2\langle \hat{z}_{m_k}^{k+1} - \tilde{z}_{m_k}^{k+1}, \tilde{z}_{m_k}^{k+1} - z^* \rangle + \left\| \tilde{z}_{m_k}^{k+1} - \hat{z}_{m_k}^{k+1} \right\|^2.
\end{aligned}
$$

Note that the function $f - f_{m_k}$ is $\delta$ - smooth (since $\|\nabla_{xx} f - \nabla_{xx} f_{m_k}\|^2 \leq \delta$, $\|\nabla_{xy} f - \nabla_{xy} f_{m_k}\|^2 \leq \delta$, $\|\nabla_{yy} f - \nabla_{yy} f_{m_k}\|^2 \leq \delta$), then

$$
\begin{aligned}
\left\|z_{m_k}^{k+1} - z^*\right\|^2 &\leq \left\|z_{m_k}^k - z^*\right\|^2 - \frac{3\gamma\mu}{2}\left\|\hat{u}_{m_k}^k - z^*\right\|^2 \\
&\quad + \frac{8\gamma\delta^2}{\mu}\left\|\tilde{u}_{m_k}^k - \hat{u}_{m_k}^k\right\|^2 + \frac{8\gamma}{\mu}\left\|\bar{F}_{m_k}^{k+1/2} - F(\tilde{u}_{m_k}^k)\right\|^2 \\
&\quad + 6\gamma^2\delta^2\left\|z_{m_k}^k - \tilde{u}_{m_k}^k\right\|^2 + 6\gamma^2\left\|\bar{F}_{m_k}^k - F(z_{m_k}^k)\right\|^2 + 6\gamma^2\left\|\bar{F}_{m_k}^{k+1/2} - F(\tilde{u}_{m_k}^k)\right\|^2 \\
&\quad + 2\left\|\tilde{u}_{m_k}^k - \hat{u}_{m_k}^k\right\|^2 - \left\|z_{m_k}^k - \hat{u}_{m_k}^k\right\|^2 \\
&\quad + \frac{4}{\gamma\mu}\left\|\tilde{u}_{m_k}^k - \hat{u}_{m_k}^k\right\|^2 + 2\langle\hat{z}_{m_k}^{k+1} - \tilde{z}_{m_k}^{k+1}, \tilde{z}_{m_k}^{k+1} - z^*\rangle + \left\|\tilde{z}_{m_k}^{k+1} - \hat{z}_{m_k}^{k+1}\right\|^2 \\
&\leq \left\|z_{m_k}^k - z^*\right\|^2 - \frac{3\gamma\mu}{2}\left\|\hat{u}_{m_k}^k - z^*\right\|^2 - (1 - 12\gamma^2\delta^2)\left\|z_{m_k}^k - \hat{u}_{m_k}^k\right\|^2 \\
&\quad + \left(2 + 12\gamma^2\delta^2 + \frac{4}{\gamma\mu} + \frac{8\gamma\delta^2}{\mu}\right)\left\|\tilde{u}_{m_k}^k - \hat{u}_{m_k}^k\right\|^2 \\
&\quad + 6\gamma^2\left\|\bar{F}_{m_k}^k - F(z_{m_k}^k)\right\|^2 + \left(6\gamma^2 + \frac{8\gamma}{\mu}\right)\left\|\bar{F}_{m_k}^{k+1/2} - F(\tilde{u}_{m_k}^k)\right\|^2 \\
&\quad + 2\langle\hat{z}_{m_k}^{k+1} - \tilde{z}_{m_k}^{k+1}, \tilde{z}_{m_k}^{k+1} - z^*\rangle + \left\|\tilde{z}_{m_k}^{k+1} - \hat{z}_{m_k}^{k+1}\right\|^2.
\end{aligned}
$$

By inequality $\|a + b\|^2 \geq \frac{2}{3}\|a\|^2 - 2\|b\|^2$, we have

$$
\begin{aligned}
\left\|z_{m_k}^{k+1} - z^*\right\|^2 &\leq (1 - \gamma\mu)\left\|z_{m_k}^k - z^*\right\|^2 - (1 - 3\gamma\mu - 12\gamma^2\delta^2)\left\|z_{m_k}^k - \hat{u}_{m_k}^k\right\|^2 \\
&\quad + \left(2 + 12\gamma^2\delta^2 + \frac{4}{\gamma\mu} + \frac{8\gamma\delta^2}{\mu}\right)\left\|\tilde{u}_{m_k}^k - \hat{u}_{m_k}^k\right\|^2 \\
&\quad + 6\gamma^2\left\|\bar{F}_{m_k}^k - F(z_{m_k}^k)\right\|^2 + \left(6\gamma^2 + \frac{8\gamma}{\mu}\right)\left\|\bar{F}_{m_k}^{k+1/2} - F(\tilde{u}_{m_k}^k)\right\|^2 \\
&\quad + 2\langle\hat{z}_{m_k}^{k+1} - \tilde{z}_{m_k}^{k+1}, \tilde{z}_{m_k}^{k+1} - z^*\rangle + \left\|\tilde{z}_{m_k}^{k+1} - \hat{z}_{m_k}^{k+1}\right\|^2.
\end{aligned}
$$

$\square$

**Lemma 8** *Let for problem* (11) *we use Extragradient method with starting point $z_{m_k}^k$ and number of iterations:*

$$
T = \mathcal{O}\left((1 + \gamma L)\log\frac{1}{\tilde{e}}\right). \tag{31}
$$

*Then for an output $\tilde{u}_{m_k}^k$ it holds that*

$$
\left\|\tilde{u}_{m_k}^k - \hat{u}_{m_k}^k\right\|^2 \leq \tilde{e}\left\|z_{m_k}^k - \hat{u}_{m_k}^k\right\|^2.
$$

**Theorem 9** *Let problem* (11) *be solved by Extragradient with precision $\tilde{e}$:*

$$
\tilde{e} = \frac{1}{2\left(2 + 12\gamma^2\delta^2 + \frac{4}{\gamma\mu} + \frac{8\gamma\delta^2}{\mu}\right)} \tag{32}
$$

*and number of iterations $T$ from* (31)*. Suppose that parameters $H_0$ and $H_1$ satisfy*

$$
\begin{aligned}
H_0 &= \mathcal{O}\left(\frac{1}{\sqrt{\rho}}\log\left(\frac{\left(\gamma^2 + \frac{\gamma}{\mu}\right) \cdot M(L\Omega + G)^2}{\varepsilon\gamma\mu}\right)\right), \\
H_1 &= \mathcal{O}\left(\frac{1}{\sqrt{\rho}}\log\left(\frac{\left(1 + \gamma^2 L^2 + \frac{\gamma L^2}{\mu}\right) \cdot M\Omega^2}{\varepsilon\gamma\mu}\right)\right)
\end{aligned} \tag{33}
$$

*Additionally, let us choose stepsize $\gamma$ as follows*

$$\gamma = \min \left\{ \frac{1}{12\mu}; \frac{1}{7\delta} \right\}. \tag{34}$$

*Then Algorithm 1 converges linearly to the solution $z^*$ and it holds that $\left\| z^K - z^* \right\|^2 \sim \varepsilon$ after*

$$K = \mathcal{O}\left( \frac{1}{\gamma\mu} \log \frac{\left\| z^0 - z^* \right\|^2}{\varepsilon} \right) \quad \text{iterations.} \tag{35}$$

**Proof:** Combining results from Lemma 7 and 8 gives

$$
\begin{aligned}
\left\| z_{m_k}^{k+1} - z^* \right\|^2 &\le (1 - \gamma\mu) \left\| z_{m_k}^k - z^* \right\|^2 - (1 - 3\gamma\mu - 12\gamma^2\delta^2) \left\| z_{m_k}^k - \hat{u}_{m_k}^k \right\|^2 \\
&\quad + \left( 2 + 12\gamma^2\delta^2 + \frac{4}{\gamma\mu} + \frac{8\gamma\delta^2}{\mu} \right) \left\| \tilde{u}_{m_k}^k - \hat{u}_{m_k}^k \right\|^2 \\
&\quad + 6\gamma^2 \left\| \bar{F}_{m_k}^k - F(z_{m_k}^k) \right\|^2 + \left( 6\gamma^2 + \frac{8\gamma}{\mu} \right) \left\| \bar{F}_{m_k}^{k+1/2} - F(\tilde{u}_{m_k}^k) \right\|^2 \\
&\quad + 2\langle \hat{z}_{m_k}^{k+1} - \tilde{z}_{m_k}^{k+1}, \tilde{z}_{m_k}^{k+1} - z^* \rangle + \left\| \tilde{z}_{m_k}^{k+1} - \hat{z}_{m_k}^{k+1} \right\|^2.
\end{aligned}
$$

With the choice $e$ from (32) and $\gamma$ from (34), we obtain

$$
\begin{aligned}
\left\| z_{m_k}^{k+1} - z^* \right\|^2 &\le (1 - \gamma\mu) \left\| z_{m_k}^k - z^* \right\|^2 \\
&\quad + 6\gamma^2 \left\| \bar{F}_{m_k}^k - F(z_{m_k}^k) \right\|^2 + \left( 6\gamma^2 + \frac{8\gamma}{\mu} \right) \left\| \bar{F}_{m_k}^{k+1/2} - F(\tilde{u}_{m_k}^k) \right\|^2 \\
&\quad + 2\langle \hat{z}_{m_k}^{k+1} - \tilde{z}_{m_k}^{k+1}, \tilde{z}_{m_k}^{k+1} - z^* \rangle + \left\| \tilde{z}_{m_k}^{k+1} - \hat{z}_{m_k}^{k+1} \right\|^2.
\end{aligned}
$$

Passing from the local $z_{m_k}^{k+1}$ and $z_{m_k}^k$ to $\bar{z}^{k+1}$ and $\bar{z}^k$, we have

$$
\begin{aligned}
\left\| \bar{z}^{k+1} - z^* \right\|^2 &\le (1 - \gamma\mu) \left\| \bar{z}^k - z^* \right\|^2 \\
&\quad + 6\gamma^2 \left\| \bar{F}_{m_k}^k - F(z_{m_k}^k) \right\|^2 + \left( 6\gamma^2 + \frac{8\gamma}{\mu} \right) \left\| \bar{F}_{m_k}^{k+1/2} - F(\tilde{u}_{m_k}^k) \right\|^2 \\
&\quad + 2\| \hat{z}_{m_k}^{k+1} - \tilde{z}_{m_k}^{k+1} \| \cdot \| \tilde{z}_{m_k}^{k+1} - z^* \| + \left\| \tilde{z}_{m_k}^{k+1} - \hat{z}_{m_k}^{k+1} \right\|^2 \\
&\quad + 2\| z_{m_k}^{k+1} - \bar{z}^{k+1} \| \cdot \| \bar{z}^{k+1} - z^* \| + \left\| z_{m_k}^{k+1} - \bar{z}^{k+1} \right\|^2 \\
&\quad + 2\| z_{m_k}^k - \bar{z}^k \| \cdot \| \bar{z}^k - z^* \| + \left\| z_{m_k}^k - \bar{z}^k \right\|^2. \tag{36}
\end{aligned}
$$

Further we will work separately only with the last 4 lines, because the last 4 lines depend on the number of iterations $H_0$ and $H_1$, then we can make them small by choosing the correct $H_0$ and $H_1$.

$$
\begin{aligned}
\text{Err}(k) &= 6\gamma^2 \left\| \bar{F}_{m_k}^k - F(z_{m_k}^k) \right\|^2 + \left( 6\gamma^2 + \frac{8\gamma}{\mu} \right) \left\| \bar{F}_{m_k}^{k+1/2} - F(\tilde{u}_{m_k}^k) \right\|^2 \\
&\quad + 2\| \hat{z}_{m_k}^{k+1} - \tilde{z}_{m_k}^{k+1} \| \cdot \| \tilde{z}_{m_k}^{k+1} - z^* \| + \left\| \tilde{z}_{m_k}^{k+1} - \hat{z}_{m_k}^{k+1} \right\|^2 \\
&\quad + 2\| z_{m_k}^{k+1} - \bar{z}^{k+1} \| \cdot \| \bar{z}^{k+1} - z^* \| + \left\| z_{m_k}^{k+1} - \bar{z}^{k+1} \right\|^2 \\
&\quad + 2\| z_{m_k}^k - \bar{z}^k \| \cdot \| \bar{z}^k - z^* \| + \left\| z_{m_k}^k - \bar{z}^k \right\|^2 \\
&\le 6\gamma^2 \left\| \bar{F}_{m_k}^k - F(z_{m_k}^k) \right\|^2 + \left( 6\gamma^2 + \frac{8\gamma}{\mu} \right) \left\| \bar{F}_{m_k}^{k+1/2} - F(\tilde{u}_{m_k}^k) \right\|^2 \\
&\quad + 2\| \hat{z}_{m_k}^{k+1} - \tilde{z}_{m_k}^{k+1} \| \cdot \Omega + \left\| \tilde{z}_{m_k}^{k+1} - \hat{z}_{m_k}^{k+1} \right\|^2 \\
&\quad + 2\| z_{m_k}^{k+1} - \bar{z}^{k+1} \| \cdot \Omega + \left\| z_{m_k}^{k+1} - \bar{z}^{k+1} \right\|^2 \\
&\quad + 2\| z_{m_k}^k - \bar{z}^k \| \cdot \Omega + \left\| z_{m_k}^k - \bar{z}^k \right\|^2
\end{aligned}
$$

Next we use the definition of $\bar{z}^k$ and $\bar{z}^{k+1}$ and the fact from line 6 of Algorithm 2: $M\tilde{z}_{m_k}^{k+1} = \sum_{i=1}^M \hat{z}_i^{k+1}$, and get

$$
\begin{aligned}
\text{Err}(k) \leq\ & 12\gamma^2 \left\| \bar{F}_{m_k}^k - \frac{1}{M}\sum_{i=1}^M F_i(z_i^k) \right\|^2 + 12\gamma^2 \left\| \frac{1}{M}\sum_{i=1}^M F_i(z_i^k) - F(z_{m_k}^k) \right\|^2 \\
& + \left(12\gamma^2 + \frac{16\gamma}{\mu}\right) \left\| \bar{F}_{m_k}^{k+1/2} - \frac{1}{M}\sum_{i=1}^M F_i(u_i^k) \right\|^2 \\
& + \left(12\gamma^2 + \frac{16\gamma}{\mu}\right) \left\| \frac{1}{M}\sum_{i=1}^M F_i(u_i^k) - F(\tilde{u}_{m_k}^k) \right\|^2 \\
& + 2\left\| \hat{z}_{m_k}^{k+1} - \frac{1}{M}\sum_{i=1}^M \hat{z}_i^{k+1} \right\| \cdot \Omega + \left\| \frac{1}{M}\sum_{i=1}^M \hat{z}_i^{k+1} - \hat{z}_{m_k}^{k+1} \right\|^2 \\
& + 2\left\| \text{proj}[\hat{z}_{m_k}^{k+1}] - \frac{1}{M}\sum_{i=1}^M \text{proj}[\hat{z}_i^{k+1}] \right\| \cdot \Omega + \left\| \text{proj}[\hat{z}_{m_k}^{k+1}] - \frac{1}{M}\sum_{i=1}^M \text{proj}[\hat{z}_i^{k+1}] \right\|^2 \\
& + 2\left\| \text{proj}[\hat{z}_{m_k}^{k}] - \frac{1}{M}\sum_{i=1}^M \text{proj}[\hat{z}_i^{k}] \right\| \cdot \Omega + \left\| \text{proj}[\hat{z}_{m_k}^{k}] - \frac{1}{M}\sum_{i=1}^M \text{proj}[\hat{z}_i^{k}] \right\|^2 \\
\leq\ & 12\gamma^2 \left\| \bar{F}_{m_k}^k - \frac{1}{M}\sum_{i=1}^M F_i(z_i^k) \right\|^2 + 12\gamma^2 \left\| \frac{1}{M}\sum_{i=1}^M F_i(z_i^k) - F(z_{m_k}^k) \right\|^2 \\
& + \left(12\gamma^2 + \frac{16\gamma}{\mu}\right) \left\| \bar{F}_{m_k}^{k+1/2} - \frac{1}{M}\sum_{i=1}^M F_i(u_i^k) \right\|^2 \\
& + \left(12\gamma^2 + \frac{16\gamma}{\mu}\right) \left\| \frac{1}{M}\sum_{i=1}^M F_i(u_i^k) - F(\tilde{u}_{m_k}^k) \right\|^2 \\
& + 2\left\| \hat{z}_{m_k}^{k+1} - \frac{1}{M}\sum_{i=1}^M \hat{z}_i^{k+1} \right\| \cdot \Omega + \left\| \frac{1}{M}\sum_{i=1}^M \hat{z}_i^{k+1} - \hat{z}_{m_k}^{k+1} \right\|^2 \\
& + \frac{2}{M}\sum_{i=1}^M \left\| \hat{z}_{m_k}^{k+1} - \hat{z}_i^{k+1} \right\| \cdot \Omega + \frac{1}{M}\sum_{i=1}^M \left\| \hat{z}_{m_k}^{k+1} - \hat{z}_i^{k+1} \right\|^2 \\
& + \frac{2}{M}\sum_{i=1}^M \left\| \hat{z}_{m_k}^{k} - \hat{z}_i^{k} \right\| \cdot \Omega + \frac{1}{M}\sum_{i=1}^M \left\| \hat{z}_{m_k}^{k} - \hat{z}_i^{k} \right\|^2 \\
\leq\ & 12\gamma^2 \left\| \bar{F}_{m_k}^k - \frac{1}{M}\sum_{i=1}^M F_i(z_i^k) \right\|^2 + \left(12\gamma^2 + \frac{16\gamma}{\mu}\right) \left\| \bar{F}_{m_k}^{k+1/2} - \frac{1}{M}\sum_{i=1}^M F_i(u_i^k) \right\|^2 \\
& + 2\Omega \left\| \hat{z}_{m_k}^{k+1} - \frac{1}{M}\sum_{i=1}^M \hat{z}_i^{k+1} \right\| + \left\| \frac{1}{M}\sum_{i=1}^M \hat{z}_i^{k+1} - \hat{z}_{m_k}^{k+1} \right\|^2 \\
& + 12\gamma^2 \frac{1}{M}\sum_{i=1}^M \left\| F_i(z_i^k) - F_i(z_{m_k}^k) \right\|^2 + \left(12\gamma^2 + \frac{16\gamma}{\mu}\right)\frac{1}{M}\sum_{i=1}^M \left\| F_i(u_i^k) - F_i(\tilde{u}_{m_k}^k) \right\|^2 \\
& + \frac{2}{M}\sum_{i=1}^M \left\| \hat{z}_{m_k}^{k+1} - \frac{1}{M}\sum_{j=1}^M \hat{z}_j^{k+1} + \frac{1}{M}\sum_{j=1}^M \hat{z}_j^{k+1} - \hat{z}_i^{k+1} \right\| \cdot \Omega \\
& + \frac{1}{M}\sum_{i=1}^M \left\| \hat{z}_{m_k}^{k+1} - \frac{1}{M}\sum_{j=1}^M \hat{z}_j^{k+1} + \frac{1}{M}\sum_{j=1}^M \hat{z}_j^{k+1} - \hat{z}_i^{k+1} \right\|^2
\end{aligned}
$$

$$+ \frac{2}{M} \sum_{i=1}^{M} \left\| \hat{z}_{m_k}^k - \frac{1}{M} \sum_{j=1}^{M} \hat{z}_j^k + \frac{1}{M} \sum_{j=1}^{M} \hat{z}_j^k - \hat{z}_i^k \right\| \cdot \Omega$$

$$+ \frac{1}{M} \sum_{i=1}^{M} \left\| \hat{z}_{m_k}^k - \frac{1}{M} \sum_{j=1}^{M} \hat{z}_j^k + \frac{1}{M} \sum_{j=1}^{M} \hat{z}_j^k - \hat{z}_i^k \right\|^2$$

$$\leq 12\gamma^2 \left\| \bar{F}_{m_k}^k - \frac{1}{M} \sum_{i=1}^{M} F_i(z_i^k) \right\|^2 + \left( 12\gamma^2 + \frac{16\gamma}{\mu} \right) \left\| \bar{F}_{m_k}^{k+1/2} - \frac{1}{M} \sum_{i=1}^{M} F_i(u_i^k) \right\|^2$$

$$+ 2\Omega \left\| \hat{z}_{m_k}^{k+1} - \frac{1}{M} \sum_{i=1}^{M} \hat{z}_i^{k+1} \right\| + \left\| \frac{1}{M} \sum_{i=1}^{M} \hat{z}_i^{k+1} - \hat{z}_{m_k}^{k+1} \right\|^2$$

$$+ 12\gamma^2 L^2 \frac{1}{M} \sum_{i=1}^{M} \left\| z_i^k - z_{m_k}^k \right\|^2 + \left( 12\gamma^2 + \frac{16\gamma}{\mu} \right) L^2 \frac{1}{M} \sum_{i=1}^{M} \left\| u_i^k - \tilde{u}_{m_k}^k \right\|^2$$

$$+ \frac{2}{M} \sum_{i=1}^{M} \left\| \hat{z}_{m_k}^{k+1} - \frac{1}{M} \sum_{j=1}^{M} \hat{z}_j^{k+1} \right\| \cdot \Omega + \frac{2}{M} \sum_{i=1}^{M} \left\| \frac{1}{M} \sum_{j=1}^{M} \hat{z}_j^{k+1} - \hat{z}_i^{k+1} \right\| \cdot \Omega$$

$$+ \frac{2}{M} \sum_{i=1}^{M} \left\| \hat{z}_{m_k}^{k+1} - \frac{1}{M} \sum_{j=1}^{M} \hat{z}_j^{k+1} \right\|^2 + \frac{2}{M} \sum_{i=1}^{M} \left\| \frac{1}{M} \sum_{j=1}^{M} \hat{z}_j^{k+1} - \hat{z}_i^{k+1} \right\|^2$$

$$+ \frac{2}{M} \sum_{i=1}^{M} \left\| \hat{z}_{m_k}^k - \frac{1}{M} \sum_{j=1}^{M} \hat{z}_j^k \right\| \cdot \Omega + \frac{2}{M} \sum_{i=1}^{M} \left\| \frac{1}{M} \sum_{j=1}^{M} \hat{z}_j^k - \hat{z}_i^k \right\| \cdot \Omega$$

$$+ \frac{2}{M} \sum_{i=1}^{M} \left\| \hat{z}_{m_k}^k - \frac{1}{M} \sum_{j=1}^{M} \hat{z}_j^k \right\|^2 + \frac{2}{M} \sum_{i=1}^{M} \left\| \frac{1}{M} \sum_{j=1}^{M} \hat{z}_j^k - \hat{z}_i^k \right\|^2 .$$

Small rearrangement gives

$$\mathrm{Err}(k) \leq 12\gamma^2 \left\| \bar{F}_{m_k}^k - \frac{1}{M} \sum_{i=1}^{M} F_i(z_i^k) \right\|^2 + \left( 12\gamma^2 + \frac{16\gamma}{\mu} \right) \left\| \bar{F}_{m_k}^{k+1/2} - \frac{1}{M} \sum_{i=1}^{M} F_i(u_i^k) \right\|^2$$

$$+ 4\Omega \left\| \hat{z}_{m_k}^{k+1} - \frac{1}{M} \sum_{i=1}^{M} \hat{z}_i^{k+1} \right\| + 4 \left\| \frac{1}{M} \sum_{i=1}^{M} \hat{z}_i^{k+1} - \hat{z}_{m_k}^{k+1} \right\|^2 + 2\Omega \left\| \hat{z}_{m_k}^k - \frac{1}{M} \sum_{j=1}^{M} \hat{z}_j^k \right\|$$

$$+ 2 \left\| \hat{z}_{m_k}^k - \frac{1}{M} \sum_{j=1}^{M} \hat{z}_j^k \right\|^2 + 12\gamma^2 L^2 \frac{1}{M} \sum_{i=1}^{M} \left\| \mathrm{proj}[\hat{z}_i^k] - \mathrm{proj}[\hat{z}_{m_k}^k] \right\|^2$$

$$+ \left( 12\gamma^2 + \frac{16\gamma}{\mu} \right) L^2 \frac{1}{M} \sum_{i=1}^{M} \left\| u_i^k - \frac{1}{M} \sum_{j=1}^{M} u_j^k + \frac{1}{M} \sum_{j=1}^{M} u_j^k - \tilde{u}_{m_k}^k \right\|^2$$

$$+ \frac{2}{M} \sum_{i=1}^{M} \left\| \frac{1}{M} \sum_{j=1}^{M} \hat{z}_j^{k+1} - \hat{z}_i^{k+1} \right\| \cdot \Omega + \frac{2}{M} \sum_{i=1}^{M} \left\| \frac{1}{M} \sum_{j=1}^{M} \hat{z}_j^{k+1} - \hat{z}_i^{k+1} \right\|^2$$

$$+ \frac{2}{M} \sum_{i=1}^{M} \left\| \frac{1}{M} \sum_{j=1}^{M} \hat{z}_j^k - \hat{z}_i^k \right\| \cdot \Omega + \frac{2}{M} \sum_{i=1}^{M} \left\| \frac{1}{M} \sum_{j=1}^{M} \hat{z}_j^k - \hat{z}_i^k \right\|^2 .$$

$$\text{Err}(k) \leq 12\gamma^2 \left\| \bar{F}_{m_k}^k - \frac{1}{M}\sum_{i=1}^{M} F_i(z_i^k) \right\|^2 + \left(12\gamma^2 + \frac{16\gamma}{\mu}\right) \left\| \bar{F}_{m_k}^{k+1/2} - \frac{1}{M}\sum_{i=1}^{M} F_i(u_i^k) \right\|^2$$

$$+ 4\Omega \left\| \hat{z}_{m_k}^{k+1} - \frac{1}{M}\sum_{i=1}^{M} \hat{z}_i^{k+1} \right\| + 4 \left\| \frac{1}{M}\sum_{i=1}^{M} \hat{z}_i^{k+1} - \hat{z}_{m_k}^{k+1} \right\|^2$$

$$+ 2\Omega \left\| \hat{z}_{m_k}^k - \frac{1}{M}\sum_{j=1}^{M} \hat{z}_j^k \right\| + 2 \left\| \hat{z}_{m_k}^k - \frac{1}{M}\sum_{j=1}^{M} \hat{z}_j^k \right\|^2$$

$$+ 12\gamma^2 L^2 \frac{1}{M}\sum_{i=1}^{M} \left\| \hat{z}_i^k - \frac{1}{M}\sum_{j=1}^{M} \hat{z}_j^k + \frac{1}{M}\sum_{j=1}^{M} \hat{z}_j^k - \hat{z}_{m_k}^k \right\|^2$$

$$+ \left(24\gamma^2 + \frac{32\gamma}{\mu}\right) L^2 \frac{1}{M}\sum_{i=1}^{M} \left\| u_i^k - \frac{1}{M}\sum_{j=1}^{M} u_j^k \right\|^2$$

$$+ \left(24\gamma^2 + \frac{32\gamma}{\mu}\right) L^2 \left\| \frac{1}{M}\sum_{j=1}^{M} u_j^k - \tilde{u}_{m_k}^k \right\|^2$$

$$+ \frac{2}{M}\sum_{i=1}^{M} \left\| \frac{1}{M}\sum_{j=1}^{M} \hat{z}_j^{k+1} - \hat{z}_i^{k+1} \right\| \cdot \Omega + \frac{2}{M}\sum_{i=1}^{M} \left\| \frac{1}{M}\sum_{j=1}^{M} \hat{z}_j^{k+1} - \hat{z}_i^{k+1} \right\|^2$$

$$+ \frac{2}{M}\sum_{i=1}^{M} \left\| \frac{1}{M}\sum_{j=1}^{M} \hat{z}_j^k - \hat{z}_i^k \right\| \cdot \Omega + \frac{2}{M}\sum_{i=1}^{M} \left\| \frac{1}{M}\sum_{j=1}^{M} \hat{z}_j^k - \hat{z}_i^k \right\|^2$$

$$\leq 12\gamma^2 \left\| \bar{F}_{m_k}^k - \frac{1}{M}\sum_{i=1}^{M} F_i(z_i^k) \right\|^2 + \left(12\gamma^2 + \frac{16\gamma}{\mu}\right) \left\| \bar{F}_{m_k}^{k+1/2} - \frac{1}{M}\sum_{i=1}^{M} F_i(u_i^k) \right\|^2$$

$$+ 4\Omega \left\| \hat{z}_{m_k}^{k+1} - \frac{1}{M}\sum_{i=1}^{M} \hat{z}_i^{k+1} \right\| + 4 \left\| \frac{1}{M}\sum_{i=1}^{M} \hat{z}_i^{k+1} - \hat{z}_{m_k}^{k+1} \right\|^2$$

$$+ 2\Omega \left\| \hat{z}_{m_k}^k - \frac{1}{M}\sum_{j=1}^{M} \hat{z}_j^k \right\| + (2 + 24\gamma^2 L^2) \left\| \hat{z}_{m_k}^k - \frac{1}{M}\sum_{j=1}^{M} \hat{z}_j^k \right\|^2$$

$$+ 24\gamma^2 L^2 \frac{1}{M}\sum_{i=1}^{M} \left\| \hat{z}_i^k - \frac{1}{M}\sum_{j=1}^{M} \hat{z}_j^k \right\|^2$$

$$+ \left(24\gamma^2 + \frac{32\gamma}{\mu}\right) L^2 \frac{1}{M}\sum_{i=1}^{M} \left\| u_i^k - \frac{1}{M}\sum_{j=1}^{M} u_j^k \right\|^2$$

$$+ \left(24\gamma^2 + \frac{32\gamma}{\mu}\right) L^2 \left\| \frac{1}{M}\sum_{j=1}^{M} u_j^k - \tilde{u}_{m_k}^k \right\|^2$$

$$+ \frac{2}{M}\sum_{i=1}^{M} \left\| \frac{1}{M}\sum_{j=1}^{M} \hat{z}_j^{k+1} - \hat{z}_i^{k+1} \right\| \cdot \Omega + \frac{2}{M}\sum_{i=1}^{M} \left\| \frac{1}{M}\sum_{j=1}^{M} \hat{z}_j^{k+1} - \hat{z}_i^{k+1} \right\|^2$$

$$+ \frac{2}{M}\sum_{i=1}^{M} \left\| \frac{1}{M}\sum_{j=1}^{M} \hat{z}_j^k - \hat{z}_i^k \right\| \cdot \Omega + \frac{2}{M}\sum_{i=1}^{M} \left\| \frac{1}{M}\sum_{j=1}^{M} \hat{z}_j^k - \hat{z}_i^k \right\|^2.$$

Now we are ready to apply `AccGossip` convergence results (([28](#), [29](#), [30](#))) to each of these terms:

$$
\begin{aligned}
\mathrm{Err}(k) \leq\ & 12\gamma^2 (1 - \sqrt{\rho})^{2H_0} \cdot 2M(L\Omega + G)^2 + \left(12\gamma^2 + \frac{16\gamma}{\mu}\right)(1 - \sqrt{\rho})^{2H_0} \cdot 2M(L\Omega + G)^2 \\
& + 4\Omega (1 - \sqrt{\rho})^{H_1} \sqrt{M}\Omega + 4(1 - \sqrt{\rho})^{2H_1} M\Omega^2 \\
& + 2\Omega (1 - \sqrt{\rho})^{H_1} \sqrt{M}\Omega + (2 + 24\gamma^2 L^2)(1 - \sqrt{\rho})^{2H_1} M\Omega^2 + 24\gamma^2 L^2 (1 - \sqrt{\rho})^{2H_1} \Omega^2 \\
& + \left(24\gamma^2 + \frac{32\gamma}{\mu}\right) L^2 (1 - \sqrt{\rho})^{2H_1} \Omega^2 + \left(24\gamma^2 + \frac{32\gamma}{\mu}\right) L^2 (1 - \sqrt{\rho})^{2H_1} M\Omega^2 \\
& + 2\Omega (1 - \sqrt{\rho})^{H_1} \sqrt{M}\Omega + 2(1 - \sqrt{\rho})^{2H_1} M\Omega^2 \\
& + 2\Omega (1 - \sqrt{\rho})^{H_1} \sqrt{M}\Omega + 2(1 - \sqrt{\rho})^{2H_1} M\Omega^2 \\
\leq\ & \left(48\gamma^2 + \frac{32\gamma}{\mu}\right) \cdot M(L\Omega + G)^2 \cdot (1 - \sqrt{\rho})^{2H_0} + 10\sqrt{M}\Omega^2 \cdot (1 - \sqrt{\rho})^{H_1} \\
& + \left(10 + 96\gamma^2 L^2 + \frac{64\gamma L^2}{\mu}\right) M\Omega^2 \cdot (1 - \sqrt{\rho})^{2H_1}.
\end{aligned}
$$

Here we also use $\Omega$ and the same trick as ([24](#)). Then one can easy check that with our $H_0$ and $H_1$ from ([33](#)) it holds $\mathrm{Err}(k) \leq \mathrm{Err} \sim \varepsilon\mu\gamma$, then with ([36](#)) we get

$$
\left\| \bar{z}^{k+1} - z^* \right\|^2 \leq (1 - \gamma\mu) \left\| \bar{z}^k - z^* \right\|^2 + \mathrm{Err}.
$$

Running the recursion, we obtain

$$
\left\| \bar{z}^K - z^* \right\|^2 \leq (1 - \gamma\mu)^K \left\| \bar{z}^0 - z^* \right\|^2 + \frac{\mathrm{Err}}{\gamma\mu},
$$

which completes the proof.

$\square$

**Remark.** In the previous theorem, we obtained convergence along the point $\bar{z}^K$. This point is virtual and is not computed by the algorithm. But in fact, all local points $z_m^K$ are also very close to $\bar{z}^K$.

**Remark.** In this case ([35](#)) dose not correspond to the number of communication rounds. To compute the number of rounds we need

$$
K \times (H_0 + H_1) = \tilde{\mathcal{O}}\left( \frac{1}{\sqrt{\rho}} \left(1 + \frac{\delta}{\mu}\right) \log \frac{\left\| z^0 - z^* \right\|^2}{\varepsilon} \right).
$$

It is also easy to estimate the total number of local iterations on server:

$$
\begin{aligned}
K \times T &= \mathcal{O}\left( \frac{1}{\gamma\mu} (1 + \gamma L) \log \frac{1}{\tilde{e}} \log \frac{\left\| z^0 - z^* \right\|^2}{\varepsilon} \right) \\
&= \mathcal{O}\left( \left(\frac{1}{\gamma\mu} + \frac{L}{\mu}\right) \log \frac{1}{\tilde{e}} \log \frac{\left\| z^0 - z^* \right\|^2}{\varepsilon} \right) \\
&= \mathcal{O}\left( \left(1 + \frac{\delta}{\mu} + \frac{L}{\mu}\right) \log \frac{1}{\tilde{e}} \log \frac{\left\| z^0 - z^* \right\|^2}{\varepsilon} \right).
\end{aligned}
$$

### B.2.2 Convex-Concave case

This case is proved similarly to Theorem 6 (convergence) and Theorem 7 (inexact consensus). We just give the statement of the theorem:

**Theorem 10** *Let problem ([11](#)) be solved by Extragradient with precision e:*

$$
e = \mathcal{O}\left( \min\left\{ \frac{\varepsilon}{\delta}; \frac{\varepsilon^2}{(L\Omega + G + \delta\Omega)^2} \right\} \right)
$$

*and number of iterations $T$:*

$$T = \mathcal{O}\left( (1 + \gamma L) \log \frac{\Omega^2}{e} \right).$$

*Suppose that parameters $H_0$ and $H_1$ satisfy*

$$H_0 = \mathcal{O}\left( \frac{1}{\sqrt{\rho}} \log \left( \frac{\left( \gamma^2 + \frac{\gamma}{\mu} \right) \cdot M (L\Omega + G)^2}{\varepsilon \gamma \mu} \right) \right),$$

$$H_1 = \mathcal{O}\left( \frac{1}{\sqrt{\rho}} \log \left( \frac{\left( 1 + \gamma^2 L^2 + \frac{\gamma L^2}{\mu} \right) \cdot M \Omega^2}{\varepsilon \gamma \mu} \right) \right).$$

*Additionally, let us choose stepsize $\gamma$ as follows*

$$\gamma = \frac{1}{4\delta}.$$

*Then it holds that $\mathrm{gap}(z_{avg}^K) \sim \varepsilon$ after*

$$K = \mathcal{O}\left( \frac{\delta \Omega^2}{\varepsilon} \right) \quad \text{iterations,}$$

*where $z_{avg}^K$ define as follows: $x_{avg}^K = \frac{1}{K} \sum_{k=0}^{K} u_x^k$, $y_{avg}^K = \frac{1}{K} \sum_{k=0}^{K} u_y^k$.*

## C   Numerical Results

The numerical experiments are run on a machine with 8 Intel Core(TM) i7-9700KF 3.60GHz CPU cores with 64GB RAM. The methods are implemented in Python 3.7 using NumPy and SciPy.

In this section, we estimate the smoothness and strong convexity parameters for objectives used in all the experiments, as well as the similarity parameter. We denote the vector with all entries equal to one as $\mathbf{1}$ and the identity matrix as $I$ (with the sizes determined by the context). Given a set of data points $X = (x_1 \ldots x_N)^\top \in \mathbb{R}^{N \times d}$ and an associated set of labels $y = (y_1 \ldots y_N)^\top \in \mathbb{R}^N$, the Robust Linear Regression problem reads

$$\min_{\|w\| \leq R_w} \max_{\|r\| \leq R_r} g(w, r) := \frac{1}{2N} \sum_{i=1}^{N} (w^T (x_i + r) - y_i)^2 + \frac{\lambda}{2} \|w\|^2 - \frac{\beta}{2} \|r\|^2.$$

Note that we need constraints on $w$ to yield the bounds for smoothness and similarity parameters (this will be described below in this section). Equivalently, $g(w, r)$ can be expressed as

$$g(w, r) = \frac{1}{2N} \left\| Xw + \mathbf{1} r^\top w - y \right\|^2 + \frac{\lambda}{2} \|w\|^2 - \frac{\beta}{2} \|r\|^2,$$

and its gradient w.r.t. $w$ and $r$ writes as

$$\nabla_w g(w, r) = \frac{1}{N} \left( X^\top X w + X^\top \mathbf{1} r^\top w - X^\top y + \mathbf{1}^\top (Xw - y) r \right) + r r^\top w + \lambda w,$$

$$\nabla_r g(w, r) = w w^\top r + \frac{1}{N} \mathbf{1}^\top (Xw - y) w - \beta r.$$

The Hessian of $g(w, r)$ w.r.t. to $w$ and $r$ are

$$\nabla_{ww}^2 g(w, r) = \frac{1}{N} \left( X^\top X + (X^\top \mathbf{1} r^\top + r \mathbf{1}^\top X) \right) + r r^\top + \lambda I,$$

$$\nabla_{wr}^2 g(w, r) = \frac{1}{N} \left( X^\top \mathbf{1} w^\top + \mathbf{1}^\top (Xw - y) I \right) + r^\top w I + r w^\top,$$

$$\nabla_{rr}^2 g(w, r) = w w^\top - \beta I.$$

We are now ready to estimate the spectrum of the Hessian taking into account the constraints on $w$ and $r$. For any $v \in \mathbb{R}^d$, we have

$$\left\|\nabla^2_{ww} g(w,r) v\right\| \leq \frac{1}{N} \lambda_{\max}(X^\top X) \|v\| + R_r^2 \|v\| + \frac{1}{N} \left\|X^\top \mathbf{1}\right\| R_r \|v\| + \frac{1}{N} \left\|r\mathbf{1}^\top X v\right\| + \lambda \|v\|$$

$$\leq \left(\frac{1}{N} \lambda_{\max}(X^\top X) + R_r^2 + \frac{2}{N} R_r \left\|X^\top \mathbf{1}\right\| + \lambda\right) \cdot \|v\| =: L^g_{ww} \|v\|,$$

$$\left\|\nabla^2_{wr} g(w,r) v\right\| \leq \frac{1}{N} \left\|X^\top \mathbf{1} w^\top r\right\| + \frac{1}{N} \left\|\mathbf{1}^\top (Xw - y) v\right\| + \left\|r^\top w v\right\| + \left\|r w^\top v\right\|$$

$$\leq \left(\frac{2}{N} \left\|X^\top \mathbf{1}\right\| R_w + \frac{1}{N} \mathbf{1}^\top y + 2 R_w R_r\right) \cdot \|v\| =: L^g_{wr} \|v\|,$$

$$\left\|\nabla^2_{rr} g(x,y) v\right\| \leq \left\|w w^\top v\right\| + \beta \|v\| \leq \left(R_w^2 + \beta\right) \cdot \|v\| =: L^g_{rr} \|v\|.$$

Therefore, we can estimate the Lipschitz constant of $\nabla g(w,r)$ as $L^g = \max(L^g_{ww}, L^g_{wr}, L^g_{rr})$.

Let us discuss the bound on the similarity parameter. Given two datasets $\left\{X \in \mathbb{R}^{N \times d}, \ y \in \mathbb{R}^N\right\}$ and $\left\{\widetilde{X} \in \mathbb{R}^{\widetilde{N} \times d}, \ \widetilde{y} \in \mathbb{R}^{\widetilde{N}}\right\}$, we define

$$\widetilde{g}(w,r) = \frac{1}{2\widetilde{N}} \left\|\widetilde{X} w + \mathbf{1} r^\top w - \widetilde{y}\right\|^2 + \frac{\lambda}{2} \|w\|^2 - \frac{\beta}{2} \|r\|^2.$$

To derive the similarity coefficient $\delta^{g,\widetilde{g}}$ between functions $g$ and $\widetilde{g}$, we separately estimate $\delta^{g,\widetilde{g}}_{ww}$, $\delta^{g,\widetilde{g}}_{wr}$ and $\delta^{g,\widetilde{g}}_{rr}$.

$$\delta^{g,\widetilde{g}}_{ww} = \lambda_{\max}\left(\frac{1}{N} X^\top X - \frac{1}{\widetilde{N}} \widetilde{X}^\top \widetilde{X}\right) + 2 \left\|\frac{1}{N} X^\top \mathbf{1} - \frac{1}{\widetilde{N}} \widetilde{X}^\top \mathbf{1}\right\| R_r,$$

$$\delta^{g,\widetilde{g}}_{wr} = 2 \left\|\frac{1}{N} X^\top \mathbf{1} - \frac{1}{\widetilde{N}} \widetilde{X}^\top \mathbf{1}\right\| R_w,$$

$$\delta^{g,\widetilde{g}}_{rr} = 0.$$

We have $\delta^{g,\widetilde{g}} = \max\{\delta^{g,\widetilde{g}}_{ww}, \delta^{g,\widetilde{g}}_{wr}\}$.

Finally, we estimate the strong convexity parameter as $\mu = \max(\lambda, \beta)$.