# OpenReview forum: "Distributed Saddle-Point Problems Under Data Similarity"
_NeurIPS.cc/2021/Conference — NeurIPS 2021 Poster_

### Official Review · Reviewer_TSma · 2021-07-12

**Rating:** 6
**Confidence:** 3

**Summary:**

This paper considers the problem of solving smooth strongly convex strongly concave saddle point problems over both centralized and decentralized networks. The main novelty is the consideration of the similarity between local functions in terms of second-order information. It provides better complexity lower bounds for both centralized and decentralized cases when the similarity of local functions is better than the smoothness of the objectives, as well as a near-optimal algorithm for the centralized case.

**Limitations And Societal Impact:**

The authors addressed them.

**Main Review:**

This work provides novel and solid complexity results for distributed SPPs under the local function similarity framework. Empirical studies are also discussed.

Major questions and concerns are:
1. The paper only discusses the empirical loss (i.e., the robust regression) example that is related to the key Assumption 2 throughout the paper, which might limit its potential scope. It is beneficial to include more examples to discuss how strong it is in practice.
2. Sometimes x and y play unsymmetric roles in SPPs, while Assumption 2 treats them equally. Can the authors comment on the potential generalization of Assumption 2 to handle x, y differently to give an even tighter analysis?
3. At Line 176, is limiting $\alpha + \beta >0$ and $\theta + \psi >0$ necessary for obtaining the lower complexity bounds? Why?
4. Algorithm 1 does not explicitly use any similarly related quantities, it is just a tighter analysis under Assumption 2 to make it optimal? Authors should provide more insights about the analysis to explain how the algorithm takes advantage of similarity assumptions. Authors should also explain what example you construct to prove complexity lower bounds in the main text.
5. The results for the decentralized case do not seem ideal as the complexity depends on $O(\log^2 \frac{1}{\epsilon})$. There are also no empirical results for this case.

Minor comments are:
1. Title in pdf file mismatches with the submission
2. Line 97, n should be d
3. Line 111, L-smooth gradient should be either L-Lipschitz gradient or L-smooth
4. Line 143 and 144, notation n is reused and can cause confusion
5. Line 183 and 185, $\phi$ should be $\psi$
6. Line 3 in Alg 3, $t+1$ should be $t$

**Time Spent Reviewing:**

5

---

> ### Author Response · Authors · 2021-08-10
> **For Reviewer TSma**
>
> We thank Reviewer TSma for review, time and the insightful comments, which will help to improve our work.
>
> We have tried to answer most of the questions and comments of all reviewers in our General comments/answers. Please read them first. Next is the answer to your review.
>
> 1) Thanks! Yes, it really matters. We've added more examples - see point 3 of our General answer.
>
> 2) This is a very interesting question. It would be good to consider it in the future work. The only problem that we see in this case is that methods that take into account the difference between $x$ and $y$ are mostly used envelopes (Catalyst or more complex), which makes such methods not the best in practice - see for example
>
> *Tianyi Lin, Chi Jin, Michael I. Jordan. Near-Optimal Algorithms for Minimax Optimization*
>
> But in theory, it's interesting! Thank you!
>
> 3) We just need $\alpha + \beta \neq 0$ and $\theta + \varphi \neq 0$. This is necessary so that both terms are not equal to zero.
>
> 4) We have partly described the idea in point 6 of our General answer. In fact, data-similarity is used implicitly when a step is made on the difference $F-F_1$ (lines 3 (i) and 3 (iii)). The question of understanding Algorithm 1 among readers is very important for us, therefore we would be glad if, in discussion with the Reviewers, we would reach some good version of a clear presentation of Algorithm 1. Therefore, if you still have a question here - you are welcome.
>
> 5) We agree. Therefore, we reflected in the Conclusion (see point 6 in our General Answer) that the decentralized method is a good future work. Decentralized experiments have been added - see point 4 of our General Answer.
>
> 6) Thanks for minors! We fixed!

---

> > ### Comment · Reviewer_TSma · 2021-09-01
> > **Thank you for your response!**
> >
> > The authors' response addressed my concerns to a certain degree. Given the new evidence of the importance of considering data similarity, as well as the examples and numerical results provided, I'm raising my score to 6.

---

### Official Review · Reviewer_J36K · 2021-07-12

**Rating:** 6
**Confidence:** 4

**Summary:**

In this paper, the authors have studied solution methods for strongly-convex-strongly-concave saddle-point problems over two types of networks, namely, master/workers networks and meshed networks. For each type of network, the authors have established lower and upper complexity bounds that depend on the problem parameters, such as the degree of similarity of the local functions and the strongly convex/concave parameter.



**Main Review:**

The paper is well written and also improves the state of the art in the field. The main weakness of the paper is that the experimental results are limited and Algorithm 3 requires global information of the network.  The detailed comments are as follows:

1. The parameter $\eta$ in Algorithm 3 depends on the second largest eigenvalue of the communication matrix $W$, i.e. $\lambda_{2}(W)$, which requires the global information of the network. Can you discuss this point and provide some distributed methods that can provide a good estimate of $\lambda_{2}(W)$?

2. In experimental results, it is preferred to study the effects of the graphs (ring, random, $k$-regular expander, etc.) on the convergence of the proposed algorithm.

3. The comparison with related work in experiments seems lacking. It is preferred to compare with the algorithms in [4] and [34]. It is also of interest to see the performance comparisons when the problem is ill-conditioned.

**Time Spent Reviewing:**

6

---

> ### Author Response · Authors · 2021-08-10
> **For Reviewer J36K**
>
> We thank Reviewer J36K for review, time and the insightful comments, which will help to improve our work.
>
> We have tried to answer most of the questions and comments of all reviewers in our General comments/answers. Please read them first. Next is the answer to your review.
>
> 1) $\lambda_2$ needs to be calculated only once. This can be done, for example, by restoring the topology of the graph on one of the nodes at the beginning of the Algorithm. Unfortunately, all the accelerated consensus procedures (that we know) use information about the eigenvalues ​​of $W$. If it is not possible to restore the topology, then one can use the non-accelerated gossip protocol.
>
> 2) We added - see point 4 in our General answer
>
> 3) In our additional experiments, we compared with the other method - Local Descent-Ascent (see point 4 in our General answer). But we can add methods from [4] and [34]. Thanks!

---

> > ### Comment · Reviewer_J36K · 2021-08-30
> > **Thank you for your response**
> >
> > Thank you for the response. My concerns have been addressed.

---

### Official Review · Reviewer_HYkt · 2021-07-13

**Rating:** 5
**Confidence:** 4

**Summary:**

This paper studies distributed (both centralized and decentralized) saddle point problems. The focus is to relate the complexity to the similarity between objective function (determined by data) on different machines, thus the complexity will be reduced when the local objectives are similar to each other.

**Limitations And Societal Impact:**

Yes.

**Main Review:**

Merits: Developed distributed algorithms that take advantage of local function similarity, reducing communication complexity compared to existing works when \delta/mu << L/mu.

Concerns:

1. My first concern is the correctness of the lower bound. In the centralized case (master/workers network),  the lower bound of O(\delta/mu \log(1/\epsilon)) in this submission is different from the one in reference [2], i.e. O(\sqrt(\delta/mu)\log(1/\epsilon)), which means the lower bound in this submission is higher (tighter) than that one when \delta/\mu > 1. What causes this difference? Is the lower bound in [2] not tight? Since reference [2] already proposed an algorithm that can hit the lower bound for a special class of problems (quadratic problem) which can satisfy the assumptions in this submission, I doubt the correctness of the lower bound in this submission. The authors may want to highlight the setting difference and proof technique difference if the lower bound in this submission is indeed correct and tighter than reference [2].

2. Theorem 1, 2 in the paper and theorem 5, 6 in the supplement material use extra assumptions f and f_m is 1-smooth and 1-Lipschitz continuous. These coefficients are critical (or even the whole point given the problems considered in this paper), thus I would highly suggest to keep them as parameters explicitly (e.g., L-smooth and G-Lipschitz), only then we can make sure the theorems are telling the right story. Also, please double check other theorems in case these assumptions are used implicitly.

3. At each iteration, full gradients are required and a subproblem has be solved accurately enough. This makes the algorithm impractical, especially considering the fact that problem scales are usually quite big in distributed scenarios.

4. In Algorithm 1, F_1 and f_1 are used, are they typos? Otherwise, what is the insight to utilize F_1 and f_1 in that way?

5. Please relate the theorems in the supplement materials more clearly to the theorems in the paper, i.e., which is which.



**Time Spent Reviewing:**

3

---

> ### Author Response · Authors · 2021-08-10
> **For Reviewer HYkt**
>
> We thank Reviewer HYkt for review, time and the insightful comments, which will help to improve our work.
>
> We have tried to answer most of the questions and comments of all reviewers in our General comments/answers. Please read them first. Next is the answer to your review.
>
> 1) We think there is some misunderstanding. Let us try to explain. The paper [2] considers the **minimization** problem; we consider a wider class of problems - **saddle point problems** (minmax). Therefore, our lower bounds are different. Minimization in this case can be considered a subclass of minmax problems. [2] and we consider different problems. Each is interesting in its own way both in theory and in practice.
>
> 2) It seems to us there won't be a big problem here. This is just normalization. In this case, we simply normalized everything to $L$. The same is done, for example, in [2].
>
> 3) Here we agree. But at the current moment in the literature, the development of data-similarity methods is only in the deterministic case - see the well-known works in point 2 of our General answer. Of course, developing stochastic methods could be a good challenge for future work. Thanks!
>
> 4) This is not a typo. Please see Section 2 of our paper. At the very beginning, the notation for $F$ and $F_m$ is given.
>
> 5) Thanks! We have already done this in the current version of the paper. This will really help better understanding.

---

> > ### Comment · Reviewer_HYkt · 2021-08-21
> > **Thank you for your response.**
> >
> > Some concerns addressed. I raise the score to 5.
> >
> > 1. Thanks.
> >
> > 2. I don't think so. Those are parameters you want to compare with the literature.
> >
> > 3. The effect of data-similarity has been broadly studied in stochastic distributed algorithms. See for example:
> >
> > [1] Karimireddy, Sai Praneeth, et al. "Scaffold: Stochastic controlled averaging for federated learning." ICML 2020.
> > [2] Yu, Hao et al. "On the linear speedup analysis of communication efficient momentum SGD for distributed non-convex optimization." ICML 2019.

---

> > > ### Author Response · Authors · 2021-08-21
> > > **About comments 2 and 3 of Reviewer HYkt's response**
> > >
> > > We thank the Reviewer for spending time checking our response and reply to us. We really appreciate the chance to be engaged in a discussion. We kindly ask the Reviewer to quickly check our short reply  below  because, from Reviewer's last comments, we understand that  we have been not very clear in some of our statements (and we apologize for that). We would appreciate if you can let us know if our reply below addresses your concerns. Thank you!
> > >
> > >
> > >
> > >
> > > **R2: Reply to 2** Referring to point 2 of the last reply, we wish to clarify that our assumption about f and f_m being  1-smooth and 1-Lipschitz continuous acts only as a normalization in the proof;  it is not a   loss of generality of the analysis and does not affect the final complexity result, which still reads
> > > $\Omega\left( \Delta\left(1 + \frac{\delta}{\mu}\right) \cdot  \log \left(\frac{\| y^*\|^2}{\varepsilon}\right)\right)$, where now $\delta$ and $\mu$ are the 'un-normalized' quantities. We remark that it is not surprising that the overall complexity does not depend on the Lipschitz constant; this is actually one of the messages of this work: if similarity is properly accounted in the design of algorithms for the SPPs, a significant reduction on the number of communications can be provably achieved with respect to that of ``similarity-oblivious" methods.  Technical details proving the correctness of the above statements are reported at the end of this reply.
> > >
> > > **R3: Reply to 3**  We understand that our reply was not very clear. What we meant by "the development of data-similarity methods is only in the deterministic case" is that we are not aware of any work that successfully exploits data similarity to provably improve communication efficiently for stochastic optimization problems, let alone, stochastic saddle-point problems. We proved in our work that statistical similarity, if properly exploited, can significantly reduce the number of communication of distributed algorithms solving deterministic SPPs; and that's the advantage  of exploiting similarity. When it comes to stochastic optimization problems, it is no longer  clear whether  this is still possible. Notice that the algorithms for stochastic optimization problems as  cited by the Reviewer are of gradient type (using stochastic gradient instances). As such, they **cannot benefit from any similarity** to reduce the overall number of communications. In fact, the results in those paper show exactly that. Whether one can design distributed algorithms for stochastic optimization problems that exploit data similarity to provably reduce the overall number of communications remains an open problem.
> > >
> > >
> > >
> > >
> > > -----------------------------------------------
> > >
> > > **2. Technical details about R2 above**
> > >
> > > Let us remove the normalization by $L$: in (11) (see Appendix), let us multiply all functions by $L$. Then the resulting functions will be  $L$-smooth (instead of $1$-smooth) and  $\tilde \mu = \mu L$-strongly-convex-strongly-concave (instead of $\mu \in (0;1]$ in the current version). Furthermore,  we will have $\tilde \delta = \delta L$-related (instead of $\delta \in [0;1]$). But it is easy to check that the dual function in Lemma 2 does not change, and thus so  the final obtained lower bounds. This means that we have the same final number of communications as reported in the submitted manuscript (in terms of the un-normalized quantities):  For any $L$, $\tilde \mu$ ($L > \mu$), $\tilde \delta$ and connected graph ... the lower bound reads
> > >
> > > $\Omega\left( \Delta\left(1 + \frac{\delta}{\mu}\right) \cdot  \log \left(\frac{\| y^*\|^2}{\varepsilon}\right)\right)$
> > >
> > > =
> > > $\Omega\left( \Delta\left(1 + \frac{\delta\cdot L}{\mu\cdot L}\right) \cdot  \log \left(\frac{\| y^*\|^2}{\varepsilon}\right)\right)$
> > >
> > >  =  $\Omega\left( \Delta\left(1 + \frac{\tilde \delta}{\tilde \mu}\right) \cdot  \log \left(\frac{\| y^*\|^2}{\varepsilon}\right)\right)$

---

### Official Review · Reviewer_VL91 · 2021-07-16

**Rating:** 6
**Confidence:** 3

**Summary:**

In this work a distributed (strongly)-convex-(strongly)-concave saddle point problem (SPP) is studied, over two different types of networks, i.e., master/worker, and meshed networks. In both cases lower bounds are derived for the number of communication rounds required  to reach an $\epsilon$ optimal solution. Differently than previous works on distributed SPP, the new analysis takes into account the similarity between the local functions. The lower communication complexity for reaching an $\epsilon$-optimal solution is shown to be in the order of $\Omega \left( \frac{\Delta \delta}{\mu} \log \left( \frac{1}{\epsilon}\right) \right)$ for a master/worker network and  $\Omega \left( \frac{ \delta}{\mu \sqrt{\rho}} \log \left( \frac{1}{\epsilon}\right) \right)$ for a meshed one; the parameter $\delta$ captures the function similarity while $\Delta, \rho$ depend on the properties of the network. Then, two algorithms are developed, one for each network type, that match these bounds, up to logarithmic factors. Finally, the proposed algorithms are evaluated on a robust linear regression problem over synthetic and real data.

**Ethical Concerns:**

There are no ethical concerns.

**Limitations And Societal Impact:**

See the Main Review Section for some of this works limitations/weak points that remain to be addressed.


**Main Review:**

Some of the positive aspects of this work are the following:

1. The authors derive novel communication complexity bounds for a setting (where the similarity between the local functions is leveraged) that has not been studied before in distributed SPPs. These new results generalize known lower bounds in the sense that the known ones offer more pessimistic estimates in the setting where function similarity is considered.

2. The algorithm class under which the lower bounds are derived is wide enough to capture many interesting distributed algorithms (e.g. using gradient or Newton updates)

3. Both types of networks, that is master/worker and meshed, are common and their study is motivated by applications. For instance, master/worker networks appear in federated learning applications.

The weak points of this work, as well as some other minor comments, are given below.

1. The utility of the derived lower bound is limited, since it is useful/interesting under the condition that $\frac{\delta}{\mu} << \frac{L}{\mu}$, where $\delta$ captures function similarity, $\mu$ is the strong convexity modulus, and $L$ is the smoothness parameter.

2. A central assumption in this work is the $\delta$-similarity of the local functions. However, the specific similarity concept defined in assumption 2 is not sufficiently motivated.  For instance, why not consider similarity in terms of gradients (rather than Hessian)?  Overall, why it is important to study the problem under the assumption of function similarity?

3. The last two motivating examples in Section 2.1 do not seem to motivate the distributed SPP problem, but rather a general SPP problem. Since this is a work about distributed SPPs I think it would be fitting to discuss why these problems are interesting in the distributed setting.

4. Comments about the experiments

    a. In the experiments Algorithm 1 is compared with the extragradient method, which is ‘‘applied directly on problem (10)’’. I presume this means that the extragradient method solves problem (10) as a non-distributed problem. Then, it is not clear how the communication rounds are defined for the extragradient algorithm, in figures 1 and 2. Also,  the authors need to explain why the comparison with a non-distributed algorithm is meaningful.

    b. There is no comparison with a distributed SPP algorithm. Thus, we cannot see if the proposed algorithm is indeed better in practice in terms of communication complexity compared with algorithms developed for the same setting. Why not perform a comparison with one of the distributed SPP methods mentioned in the ‘Introduction’ section?

    c. It is mentioned in the text that the experiments over meshed networks are located in the supplementary material. However, I cannot find these results. Are they missing?

   d. What is ‘sliding’ in the legend of figure 1? Also, the legends in figures 1 and 2 are different, although according to the description both experiments compare the same algorithms.

  e. It is mentioned that ‘We solve the problem over a master/workers topology; we consider networks with 10 and 15 workers’. However, in the figures 1 and 2 the numbers are 10 and 20, respectively.

5. In the four theorems of the main text the solution $z^{\ast}$ is not defined.

**Time Spent Reviewing:**

5-10

---

> ### Author Response · Authors · 2021-08-10
> **For Reviewer VL91**
>
> We thank Reviewer VL91 for review, time and the insightful comments, which will help to improve our work.
>
> We have tried to answer most of the questions and comments of all reviewers in our General comments/answers. Please read them first. Next is the answer to your review.
>
> 1) You are right, indeed we can get improvements only when we have statistical similarity. This is a key assumption. But it is quite popular in literature and in practice - see points 1-2 of our general answer.
>
> 2) The similarity of the Hessians comes from statistics, see
>
> *Joel A. Tropp. An introduction to matrix concentration inequalities*
>
> or (15) in
>
> *Hadrien Hendrikx, Lin Xiao, Sebastien Bubeck, Francis Bach, Laurent Massoulie. Statistically Preconditioned Accelerated Gradient Method for Distributed Optimization*
>
> In fact, the similarity condition is quite often met in practice, and the similarity methods (see literature from point 2 of our General Answer) show themselves well on real problems.
>
> 3) We rewrote it in more details in point 3 in our General answer
>
> 4) а) There is a slight misunderstanding here. We of course compare the distributed extragradient method, i.e. agents compute the local gradients, send them to the server, the server takes a step, and then sends the new point back to agents.
>
> b) We added experiments with Local Gradient Descent-Ascent - see point 4 in our General answer
>
> c) We added- see point 4 in our General answer
>
> d) Thanks! It is a typo. Sliding = Algorithm 1. We fixed!
>
> e) Thanks! It is also typo! We fixed! The right is "10 and 20".
>
> 5) Thanks! We fixed!

---

> > ### Comment · Reviewer_VL91 · 2021-08-23
> > **Response to authors**
> >
> > I would like to thank the authors for responding to my comments. The weaknesses I mentioned (e.g. weak experiments and motivation) seem to be addressed. As a result, I raise my rating to 6.
> >
> > Comments:
> >
> > 4)a) OK, thank you. It is clear now. Perhaps, the addition of a brief explanation of how distributed extragradient is applied in your problem will help avoid similar misunderstandings by the readers.

---

### Author Response · Authors · 2021-08-10
**General comments/answer**

We thank all the Reviewers for the insightful comments, which will help to improve our work.

We begin  providing some general comments addressing common questions/recommendations from all the Reviewers along with a list of the major changes that will be implemented in the revised paper. Replies to specific questions of each Referee will follow under their comments.


**1) Significance of the results:** Communications are the bottleneck in most distributed systems (in many cases, communication time takes 50-90% of the total algorithm time). Exploiting data similarity in the algorithmic design can bring a significant communication saving, as it is proved to reduce significantly the overall number of communications with respect to algorithms that do not do so (and use the same information). For such algorithms, very often the number of communications is $O(1)$.

**2) On the importance of statistical similarity:** Developing solution methods for minimization problems (not in the saddle point form)  over star networks exploiting data similarity has received significant attention in the literature; examples include

*Ohad Shamir, Nati Srebro, Tong Zhang. Communication-Efficient Distributed Optimization using an Approximate Newton-type Method - ICML 2014*

*Yuchen Zhang, Xiao Lin. DiSCO: Distributed Optimization for Self-Concordant Empirical Loss - ICML 2015*

*Yossi Arjevani, Ohad Shamir. Communication complexity of distributed convex learning and optimization - NIPS 2015*

*Sashank J. Reddi, Jakub Konecny, Peter Richtárik, Barnabás Póczós, and Alex Smola. AIDE: Fast and communication efficient distributed optimization - arXiv:1608.06879*

*Shusen Wang, Farbod Roosta-Khorasani, Peng Xu, and Michael W Mahoney. GIANT: Globally improved approximate Newton method for distributed optimization - NIPS 2018*

*Hadrien Hendrikx, Lin Xiao, Sebastien Bubeck, Francis Bach, Laurent Massoulie. Statistically Preconditioned Accelerated Gradient Method for Distributed Optimization - ICML 2020*

*Amir Daneshmand, Gesualdo Scutari, Pavel Dvurechensky, Alexander Gasnikov. Newton Method over Networks is Fast up to the Statistical Precision - ICML 2021*

Recently there has been an interest also in extending these methods over general network topologies. Examples include:

*Y Sun, A Daneshmand, G Scutari, Distributed Optimization Based on Gradient-tracking Revisited: Enhancing Convergence Rate via Surrogation*

*B. Li, S. Cen, Y. Chen, and Y. Chi, Communication-efficient distributed optimization in networks with gradient tracking and variance reduction, JMLR 2020*

All the examples above  provide solid evidence of the popularity that methods exploiting data-similarity have received, theoretically and practically. We foresee a persistent interest in this direction, as optimal methods (i.e., matching lower bounds) under similarity have not been discovered yet.

Our work goes beyond classical optimization problems and derives lower complexity bounds and optimal distributed algorithms for the more general class of SPP over either star-networks or general networks, which has been an open problem for years.


We hope we clarified the importance of exploiting data-similarity as well as the significance of our contribution, theoretically and practically.


**3) More applications of distributed SPP:** We will provide more examples of practical distributed saddle point problems where similarity can be used. Specifically, we will add the following:

*a) Online transport or Wasserstein Barycenter (WB) problem* can be rewritten as saddle point problem, see

*Dvinskikh D., Tiapkin D. Improved complexity bounds in wasserstein barycenter problem - AISTATS 2020*

This representation comes from the dual view on transportation polytope. Since the data (histograms / pictures of brains) typically weighs a lot we need distributed computations. And if we split data uniformly, we will have similarity on nodes.

*b) Lagrangian based optimization:* Let us consider the distributed convex minimization problem with constraints. It can be rewritten as a saddle-point problem using Lagrangian multipliers. It is easy to check that if the original functions $f_i$ is $\delta$-related, then the new saddle-point problem is also $\delta$-related.


**4) Additional experiments.** At the moment, we are ready to provide additional experiments for the paper.
More specifically, the new results are the following:

*a) Star networks:* in addition to  the distributed extragradient method, we compare the proposed algorithm with Local Descent- Ascent from the paper (because local technique is also used for reducing number of communications):

*Yuyang Deng, Mehrdad Mahdavi, Local Stochastic Gradient Descent Ascent: Convergence Analysis and Communication Efficiency*

*b) General graph topologies:* we add experiments testing our algorithms on several different graph topologies, such as click, ring, and expander graphs. We also compare with the decentralized methods for saddles from the following works:

*Mingrui Liu, Wei Zhang, Youssef Mroueh, Xiaodong Cui, Jerret Ross, Tianbao Yang, Payel Das, A Decentralized Parallel Algorithm for Training Generative Adversarial Nets*

*Soham Mukherjee and Mrityunjoy Chakraborty.   A decentralized algorithm for large scale min-max problems*

We have created an anonymous github repository, where we uploaded the graphs of new experiments. If there is such an opportunity and permission from AC, we are ready to provide them.

**5) Conclusions.** We added a small conclusion for our paper. Here the text of the conclusion:

*We studied distristributed SPPs over networks, under data similarity. Such problems arise naturally from many applications, including machine learning and signal processing. We first derived lower complexity bounds for such problems for solution methods implementable either on star-networks or on general topologies (modeled as undirected, static graphs). These algorithms are optimal, in the sense that they achieve the lower bounds, up to log factors. The implementation of the proposed method over general network, however, is improvable: by selecting only one agent at time performing the updates, it does not fully exploit the potential computational speedup offered by the parallelism of the networking  setting. Also, the use of gossip protocols to propagate the updates of a single agent across the entire  network seems to be not very efficient. Finally, it would be interesting to combine the proposed methods with variance reduction techniques to alleviate the cost of local gradient computations. Designing distributed algorithms for SPPs overcoming these  limitations will be the subject of our future research.*

**6)Intuition and sketch of proofs.** We added a little intuition of methods and sketches of proofs to the main part of the work.

For lower bounds:

*The idea behind getting distributed lower bounds is to give an example of “bad” functions and “bad” layouts on agents. Let us consider the following functions:*

*(see (11) in supplementary materials)*

*It is easy to verify that these functions are $1$-smooth, $mu$-strongly-convex-strongly concave, and $\delta$-related. To obtain centralized bounds, let us partition function 1 into one third of agents, function 2 -- into one third, and function 3 --  into the remaining third. The solution to the general problem is nonzero in all components. But if we take the starting vector to be zero, then we can notice that the agents, without communications, will not be able to achieve non-zero components in the final output, no matter what algorithm (satisfying Definition 1) we use. Then it holds*

*(Theorem 1)*

*For the decentralized case the technique is close, but we need to consider a linear graph and use the trick with the characteristic number from [35].*

*(Theorem 2)*

For Algorithm 1:

*Briefly, Algorithm 1 combines two ideas: statistically preconditioned Bregman gradient method and extragradient. In fact, in Algorithm 1 it is difficult to see the extragradient in its classical form, the only thing that indicates this is the presence of line 3 (iii).*

Sketch of proof for Algorithm 1:

*The next lemma gives the convergence of one step*

*(Lemma 4 from supplementary materials)*

*Then, if we choose extragradient as a method for the inner saddle problem, it holds that*

*(Lemma 5 from supplementary materials)*

*Finally, the following theorem gives the parameters of Algorithm 1 and the communication complexity:*

*(Theorem 7 from supplementary materials)*

---

### Decision · Program_Chairs · 2021-09-28

**Decision:**

Accept (Poster)

**Comment:**

In this work a distributed (strongly)-convex-(strongly)-concave saddle point problem (SPP) is studied. The authors provide communication lower bounds to reach an  $\epsilon$-optimal solution. Differently than previous works on distributed SPP, the result in this work takes into account the similarity between the local functions. Further, algorithms are developed which  match the proposed lower bounds. Finally, the proposed algorithms are evaluated on a robust linear regression problem over synthetic and real data.

Pros:

1) The authors derive communication complexity bounds for the SSP setting, which appears to generalize known lower bounds for the same class of saddle point problems, but without taking the similarity into consideration.

2) The authors developed two algorithms, one for the case with central controller, and one for the case without central controller, for the considered problems. Further, the authors developed analysis for the proposed algorithms, and show that the lower bound derived can be in fact met.

Cons:

1) The intuition behind algorithm design is not clear; nor is it clear whether a few design choices is necessary for not. For example, in Algorithm 1, why instead of performing a single gradient descent step, one needs to do step (iii) combined with step (i). Consider the simple case where there is no constraint, and f_1 = 0, then it seems that step (iii) does not provide any benefit in addition to step (i). As some reviewers have pointed out, it is difficult to understand a few design choices of the proposed algorithm.

2) There is a lingering issue about how the lower bounds depends on how the fact that the authors' assumption about f and f_m being 1-smooth and 1-Lipschitz continuous (instead of having constants explicitly written down) will affect the proof.  This is an important question. The authors provided answers to it, but this requires the current proof to be changed.

3) A question related to 2)  is that when constructing lower bound, it has assumed that the function is 1-Lipschitz continuous, while everywhere in the paper, such condition is not considered. Typically this will not pose any issue since the upper bound (for a wider class of problem without assuming Lipschitz-continuous function) matches the lower bound (constructed using a smaller class). However, since the dependency on the Lipschitz constant is not clear, and the key here is to understand whether the upper bound is tight in \delta, there could be an issue. Assume that the constructed problem is L-Lipschitz. It could be that the lower bound depends on \delta and L through $\frac{\delta}{L}$. In this case, the larger the L, the weaker the dependency on \delta. However, the upper bound does not quantify such a dependency, since there is no assumption on the Lipschitzness of the function. My point is that, it is important to explicitly write down the dependency on the Lipschitz constant, and the gradient Lipschitz constant, especially when the constructed problem requires additional assumption.

3) As a minor comment, the algorithm itself is not particularly designed for problems with similarity. The similarity constant appears when providing the analysis, and in particular, when setting the accuracy for solving local problems.

Overall, I think the paper is interesting and contains good results. However, there are still a few places,  that require some substantial revision.  I believe that this paper will be made much better if these issues are addressed.



**Consistency Experiment:**

NeurIPS has a long history of experimentation. In 2014, NeurIPS ran an experiment in which 10% of submissions were reviewed by two independent committees to quantify the randomness in the review process. This year, we repeated a variant of this experiment to see how the quality of the review process has changed over time.  This paper was part of the experiment and was therefore assigned to two committees (consisting of reviewers, an Area Chair, and a Senior Area Chair) that reached independent decisions.  If both committees made the same recommendation, this recommendation was followed. If a single committee recommended acceptance, the paper was accepted (with the exception of a few cases in which the other committee identified what we considered a fatal flaw, e.g., an error in a key result).

This copy’s committee reached the following decision: **Reject**

The other committee assigned to the paper recommended **Accept (Poster)**.  You can find the other set of reviews, along with any follow up discussion with the authors here:
https://openreview.net/forum?id=dJcUhDVu1G